# A Systematic, Automated Approach for River Segmentation Tested on the Magdalena River (Colombia) and the Baker River (Chile)

Andrea Nardini [1,2,*] , Santiago Yépez [3,4,*] , Bruno Mazzorana [5], Héctor Ulloa [5], María Dolores Bejarano [6] and Alain Laraque [7]

1   Fundación CREACUA (Centro para la Recuperación de Ecosistemas ACUáticos), 40001 Riohacha, La Guajira, Colombia
2   PhD program in Advanced Forestry Engineering, E.T.S.I. Montes, Forestal y Medio Natural, Universidad Politecnica de Madrid—UPM, 28040 Madrid, Spain
3   Departamento Manejo de Bosques y Medio Ambiente, Facultad de Ciencias Forestales, Universidad de Concepción—UdeC, Concepción 407374, Chile
4   Laboratorio de Geotecnologías y Modelamiento de Recursos Naturales (LGM), Universidad de Concepción —UdeC, Concepción 407374, Chile
5   Instituto de Ciencias de la Tierra, Facultad de Ciencias, Universidad Austral de Chile, Valdivia 5090000, Chile; bruno.mazzorana@uach.cl (B.M.); ulloacontreras@gmail.com (H.U.)
6   Departamento Sistemas y Recursos Naturales, E.T.S.I. Montes, Forestal y Medio Natural, Universidad Politécnica de Madrid—UPM, 28040 Madrid, Spain; mariadolores.bejarano@upm.es
7   IRD, GET-UMR CNRS/IRD/UPS—UMR 5562 du CNRS, UMR 234 de l'IRD, 900 rue J.F. Breton, 34090 Montpellier, France; alain.laraque@ird.fr
*   Correspondence: nardiniok@gmail.com (A.N.); syepez@udec.cl (S.Y.)

**Abstract:** This paper proposes a systematic procedure to identify river reaches from a geomorphic point of view. Their identification traditionally relies on a subjective synthesis of multi-dimensional information (e.g., changes of slope, changes of width of valley bottom). We point out that some of the attributes adopted to describe geomorphic characters of a river (in particular sinuosity and confinement) depend on the length of reaches, while these latter are not yet identified; this is a source of ambiguity and introduces, at least conceptually, an unpleasant, implicit, iterative procedure. We introduce a new method which avoids this difficulty. Furthermore, it is simple, objective, and explicitly defined, and as such, it is automatable. The method requires to define and determine a set of intensive attributes, i.e., attributes that are independent of the segment length. The reaches are then identified by the intersection of the segmentations induced by such attributes. We applied the proposed procedure in two case studies, the Magdalena River (Colombia) and the Baker River (Chile), and investigated whether the adoption of the traditional approach for the definition of reaches would lead to a different result. We conclude that there would be no detectable differences. As such, the method can be considered an improvement in geomorphic river characterization.

**Keywords:** river reaches; segmentation; automated approach; fluvial geomorphology; River Styles

## 1. Introduction

The key issue addressed in this paper is the identification of river reaches from a geomorphic point of view. In principle, one may describe the character of a river all along its course, without segmenting it. Indeed, a river is a composite, often complex pattern of continuous transitions from source to sink. Nonetheless, it is very useful to identify parts of the river, clearly distinguishable from one another, to develop a meaningful description of the character and behavior of the river and to design

proper interventions. This is where the concept of "reach" comes into play. "River reach", as noted by Parker et al. [1], is a commonly used concept, though its definition is quite loose and non-homogenous amongst authors (Kellerhals et al. [2] is the key starting source; also see, more recently, Gurnell et al. [3]). In this paper, we define the reach to be a length of the river that presents a distinctive and recognizable geomorphic character. This implies that every single descriptor (like, for instance, the local active channel width), can be given a meaningful, unique, representative value for the reach itself. In what follows, we name such descriptors "attributes". Within a reach, the variable underlying each attribute may vary even significantly, but it maintains a recognizable, prevailing spatial pattern.

Different approaches to identify reaches have been explored in the literature, which are discussed in detail in Section 2 by enlightening their key strengths and weaknesses. The approach we propose, presented in Section 3, overcomes these weaknesses, while being extremely simple. It just requires identifying a core set of intensive attributes (i.e., independent on a segment length), each one implying a specific, independent segmentation of the river. Then, the reaches are identified by the intersection of such segmentations. Extensive and binary attributes (e.g., confinement, presence/absence of geomorphological units) are then computed as statistics over the reaches just identified. Hence, they do not contribute to defining the reaches, but they still play an important role in distinguishing different typologies of reaches, together with the main attributes.

In Section 4, we apply our new procedure to the Magdalena River in Colombia and the Baker River in Chile. The two cases are very diverse in terms of environments and river typologies. Herein, we test our approach by comparing its output with the one obtained through a manual procedure based on expert judgment. Section 5 clarifies the novelty and added value of the proposed method.

Finally, we draw our conclusions in Section 6.

## 2. Previous Approaches to River Segmentation

In this section, we discuss the known approaches to river segmentation, namely:

- Manual segmentation based on expert judgment, the most applied one (see Section 2.1).
- Artificial Intelligence and Machine Learning algorithms based on image recognition (see Section 2.2).
- Statistical algorithms (see Section 2.3).
- Logical or heuristic algorithms (see Section 2.4).

Mixed solutions are also possible, but it is out of the scope of this paper to discuss all possibilities exhaustively.

### 2.1. Manual Segmentation Based on Expert Judgement

As summarized for instance in the REFORM project (Bizzi et al. [4]), manual segmentation is the traditional approach to determine reaches. That procedure starts from a preliminary segmentation (based on changes of slope, valley bottom (VB) width, basin area because of tributaries' input). This segmentation is further refined at a more detailed scale by considering confinement, sinuosity, planform, and additional aspects, amongst which the presence and character of geomorphic units (as in the River Styles Framework of Brierley and Fryirs [5]) through a holistic recognition by experts (a structural, taxonomic approach for defining such units was proposed by Wheaton et al. [6]).

There are several issues associated with this manual procedure. The subjectivity implied by expert judgment can add flexibility and capability to appraise things that really matter. However, on the one hand, being subjective, it is not replicable, and it is time consuming. More importantly, manual segmentation includes a kind of intrinsic paradox: several attributes (e.g., sinuosity, confinement), denoted as extensive attributes, depend on the segment length chosen (the reach), a length which at the beginning is still undefined, being the reach itself the sought output. In other words, the characterization is virtually impossible without the reaches, yet these can only be defined through a characterization process (see Figure 1 for visual examples). The only practical way out is to undertake iterations that are,

however, traditionally driven by intuition and hence remain obscure or simply unexplored. Therefore, each expert may come up with a different segmentation for the same river.

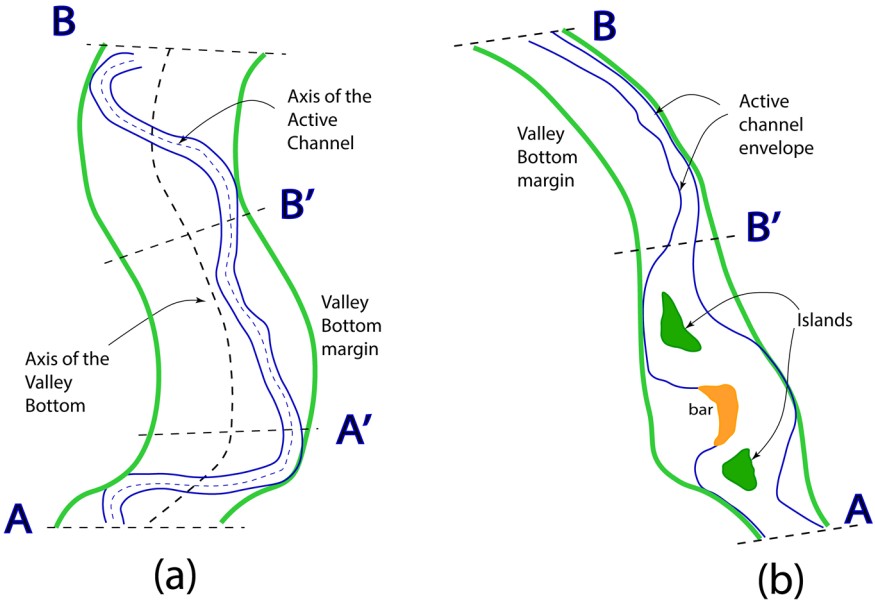

**Figure 1.** Examples of ambiguity related to extensive attributes. (**a**) Sinuosity of segment A-B is much higher than that of segment A'-B' which is almost straight; that of A-A' is even higher; (**b**) Regarding confinement, the whole segment A-B is identified as "partly confined" because along its length more than 10%, but less than 85% of either banks of the active channel is in contact with the VB margin; however, based on planform, two well different sub segments are identified (A-B'; B'-B) and confinement for segment A-B' gets the value "confined" (because the total contact length is more than 85%) (definition of confinement according to Brierley and Fryirs, [5]; Fryirs et al. [7]; O'Brien et al. [8]). A similar behavior occurs with other attributes (those concerning the presence of geomorphic units): each time an attribute comes into play, and a further segmentation is introduced, all previously assessed attributes need to be re-assessed possibly producing different values. In addition, when the channel migrates (adjustment without changing its character), "confinement" may possibly change; since this is the starting point of the River Styles procedural tree, this may produce dramatic changes in the classification and hence significant re-segmentation work.

### 2.2. Artificial Intelligence and Machine Learning Algorithms Based on Image Recognition

Artificial Intelligence-AI developed algorithms potentially suitable for our problem (notably those named Machine Learning based on Artificial Neural Networks: Rutkowski, [9]; Olson and Delen [10]; Buscombe and Ritchie [11]; Reichstein et al. [12]; Tsagkatakis et al. [13]; Yuan et al. [14]). These algorithms go beyond the pure interpretation of the spatial configuration (which is itself very powerful, as shown, for instance, by Jacquez, [15]), being able to interpret an assemblage of a wide array of elements like spatial configuration, altimetry and "colors" (i.e., based on the specific characteristics of each material according to the intensity of light reflected at different wavelengths) through image comparison. Deep convolutional neural networks are indeed particularly promising for image classification (Krizhevsky et al. [16], Indolia et al. [17]).

Nevertheless, this approach still faces some hindrances which are schematically pointed out here. A Neural Network depends on training and facing new situations of which it has no yet experience may be challenging. Training requires in general a large amount of information that in the "river" case should be prevailingly provided manually, which could constitute a too heavy and time-consuming task (Goodfellow [18] recommends 5000 thousand samples per class!). Bhamare and Suryawanshi [19] indeed stated that "The general problem of recognizing complex, arbitrary patterns with arbitrary orientation, location, and scale, remains unsolved . . . ". A scale of analysis must be specified in AI

algorithms to avoid they "get lost" jumping into finer and finer details, while looking for irrelevant distinctions of form assemblages. Clearly, the outputs of AI algorithms heavily depend on images resolution and are hardly exportable to other settings. Finally, even if AI can potentially recognize stretches with similar patterns of characteristics, assigning them a given label, AI can hardly explicitly inform on the role of the attributes in leading to such a determination; advances exist, however, that go in this direction (e.g., Toms et al. [20]).

## 2.3. Statistical Algorithms

Statistical algorithms aim at obtaining a holistic synthesis starting from reductionist, analytical information related to the river axis. More precisely, they need a regular discretization of the river (typically the active channel envelope) into "slices", i.e., short polygons whose main axis is perpendicular to the VB axis. A value of each one of the relevant attributes, or the information needed to determine them, is associated with each slice (e.g., the presence/absence of a geomorphic unit, see Figure 2). We refer to this base as the "skeleton", a digital structure which can be set up for instance by means of the Fluvial Corridor Toolbox (FCT, Roux et al. [21]).

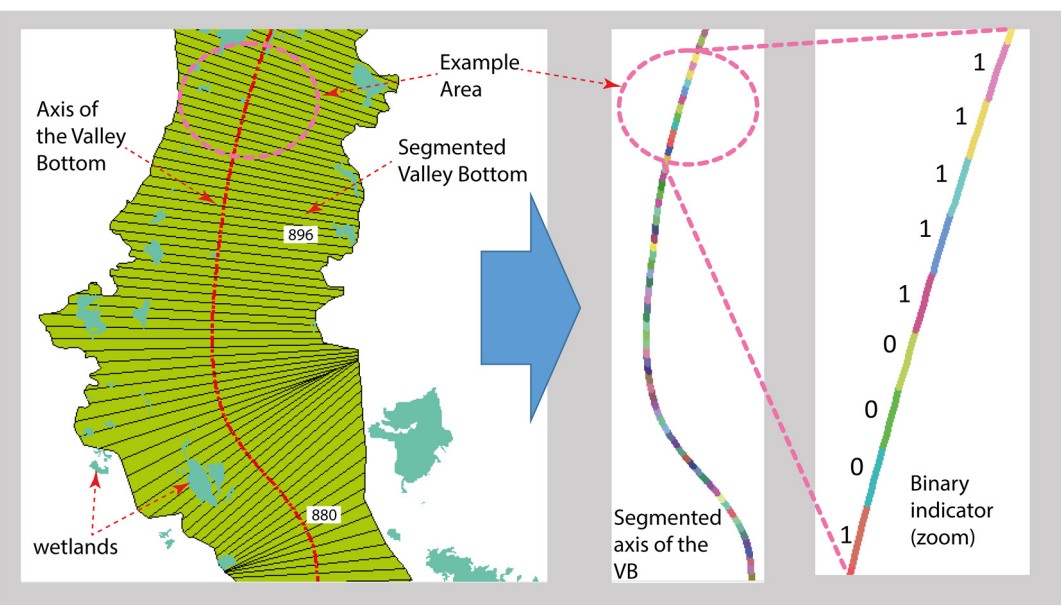

**Figure 2.** Creating the "river skeleton": geomorphological units in the active channel (e.g., islands, bars, pools, riffle) or in the VB (e.g., levees, ridge and swales, paleo channels, wetlands) translate into a binary attribute of presence (1) or absence (0) in each discretization element (either of the floodplain or of the active channels envelope) along the VB axis. The example here refers to wetlands within the floodplain (warm green), represented by the cold green spots; notice that some are external to the VB because they lie at a higher elevation (hence not linked to the river). The right-hand figure represents the final binary information (here related just to the segment within the circle).

For mono-dimensional attributes, statistical approaches that automatically and objectively identify relevant differences have been discussed by Parker et al. [1], who also provide an interesting literature overview starting from geological analysis (Clifford and Harmar, [22]), recalling Davis' [23] classification in: (i) "local boundary hunting" and (ii) "global zonation". The global zonation algorithms are best suited to identify river reach boundaries because they statistically minimize within-reach variation and maximize between-reach differences and therefore are less influenced by local inconsistencies in the data sequence.

Hubert [24] introduced a statistical test to define reaches which works in a similar manner; it is currently considered the key 'tool' to support reaches' identification within the Fluvial Corridor Toolbox

(Roux et al. [21]). However, the output of Hubert's test (stretches of different colors in Figure 3) can be questionable. For instance, the stretch marked by the two red segments in Figure 3 would certainly be classified by expert judgment as narrower with respect to the one above it (notice that there is no criterion linked to the length as Hubert test determined both short and long segments) (source: Nardini et al. [25]).

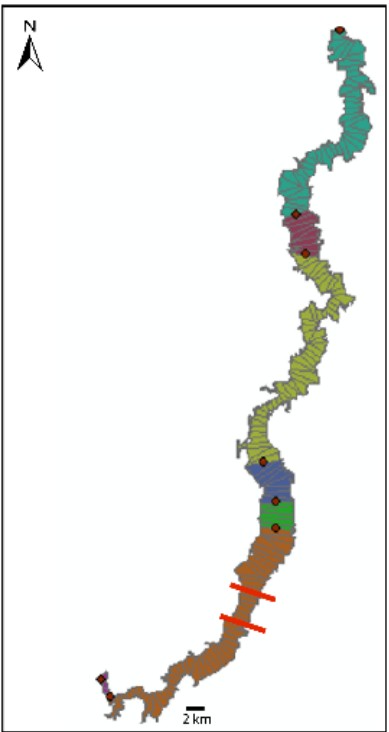

**Figure 3.** Example of segmentation of a continuous, quantitative attribute (the VB width of a stretch of the Magdalena River) via Hubert statistical test as implemented in the Fluvial Corridor Toolbox: the output (stretches of different colors) is questionable (see text).

Martinez-Fernandez et al. [26] applied similar tests by comparing three different techniques: the Pettitt test, the Mann–Kendall test and the non-parametric Multi Response Permutation Procedure (MRPP) test by using three different river attributes (VB width, active channel width, slope). Their conclusion is that the three tests lead to quite different results, Pettitt and MRPP being the most similar. Incidentally, they find that the VB width is the most explaining attribute.

It must be recognized that all these techniques have been designed taking into account quantitative, continuous, mono-dimensional variables, but, when binary (e.g., presence/absence of a geomorphic unit) or categorical variables (e.g., planform typology) are involved, they conceptually and practically fail, as it can be confirmed by direct testing.

We are not aware of working extensions to multivariate data sequences, as envisaged for instance by Davis, [23]. The MRPP test may seem to be a statistical algorithms-based approach; however, it better fits into a logical or heuristic category (see Section 2.4). In fact, in its operational version, it resembles a randomized search more than to any statistical algorithm because it randomly considers clusters of sequential slices (i.e., by setting "cuts" of the river axis) and measures each time the dispersion around the mean in each cluster, looking for a choice of clusters that minimizes it, while maximizing the difference amongst clusters. Unfortunately, MRPP cannot work with categorical variables like, for instance, planform typology.

## 2.4. Logical or Heuristic Algorithms

The main representative within this category is the family of clustering algorithms. For the multi-dimensional case, Bizzi and Lerner [27] present an interesting development based on the Self Organizing Maps (SOM) technique, which can be classified as a logical or heuristic type algorithm, not being based on statistical criteria. It groups multivariate data in an exogenously imposed number N of clusters; such data, once grouped, can be tracked back by their identifier, and located along the river axis, so that reaches can be identified. Notice that here, again, an arbitrary, a priori, regularly spaced discretization ("slices") needs to be carried out and all attributes have to be assessed on each such "slices", as shown in Figure 2.

The purpose of SOM is to group data which are similar; in addition, it is inherently multidimensional. Therefore, it would be reasonable to apply it to cluster data by considering all the relevant attributes at a time, i.e., all their dimensions (incidentally, Bizzi and Lerner [27] considered a limited set of attributes which do not coincide with those suited for a River Styles analysis). In such a case, however, difficulties arise in the presence of binary or extensive attributes because these, by constitution, cannot be meaningfully determined over arbitrary segments. Indeed, different discretizations would produce data sets with completely different behaviors: for instance, a binary variable signaling the presence of islands can occur with value 1 in one segment and value 0 in the following segment, while both segments actually belong to a given longer reach with sporadic islands. Moreover, SOM cannot deal with categorical variables because it relies, as a measure of "similitude", on an Euclidean distance which can only be computed for quantitative variables (or at least for variables defined on interval scales, where differences have a meaning, as explained in Volta and Servida [28]).

## 3. A New Approach to Identify "Reaches" and Their Characters

The problem of defining a reach lies at the basis of geomorphic river analysis and it pervades the River Styles Framework (Brierley and Fryirs [5]), hereafter denoted RSF. Indeed, the identification and classification of reaches is based there on a set of attributes most of which are extensive (i.e., length depending), leading to the paradox already discussed in Section 2.1. In this section, we propose a new approach and in what follows we often refer to the RSF, as it offers a consistent context to illustrate some of the key ideas. Accordingly, "confinement", unless differently specified, is adopted in the sense defined by Fryirs et al. [7] and O'Brien et al. [8], i.e., as a statistical measure of the contact length between the active channel and either of the VB margins.

The identification of geomorphic units (such as active channels, islands, bars) with their characters and descriptors of the fluvial elements can be achieved via analysis and interpretation of remote sensed data (Yepez et al. [29]). We assume that this task can be (or will soon be) accomplished in an automated fashion, as done for example by Bertrand et al. [30] or De Marchi et al. [31]. Here, we build on this ability and we go a step further: we identify river reaches by interpreting the data assemblage so obtained through suitable attributes. We do not intend to classify the reaches, an exercise that can be conducted for instance through the RSF; we just want to identify reaches. The attributes are always associated with the river axis, they are mono-dimensional and may vary along the river, though only in a discrete fashion (i.e., they are defined over a segmentation). Attributes can be based either on mono-dimensional information or on multi-dimensional information (e.g., planform typology—a mono dimensional, categorical attribute—is specified based on several features, like number of channels, presence of bars, etc.). The variable underlying each attribute may vary continuously along the river (e.g., active channel width or average size of sediment grains within a cross section), or in a discrete fashion (for instance, presence/absence of islands is a binary variable). We refer to all this information as "reductionist information" in order to stress that it tells us something just on each specific piece of territory, yet it does not provide an overview.

From this information, we want to obtain a holistic synthesis leading to a meaningful segmentation of the river by capturing a prevailing character of the river on an appropriate scale. The scale can depend on the river at hand. For example, the presence of one or more islands along a stretch of the

river may or may not be sufficient to identify that portion as an "island reach", as it depends on the river size and the stretch length.

Our idea is implemented in three steps (see Figure 4):

- To choose and assess core geomorphic attributes (Section 3.1);
- To identify geomorphic reaches as a "least common denominator" (spatial GIS intersection) of such segmentations (Section 3.2);
- To refine the results obtained by considering anomalies (like artificial disconnections due to civil works) and by neglecting (possibly in an automated way) the separation of very short reaches with respect to the scale of analysis (Section 3.3).

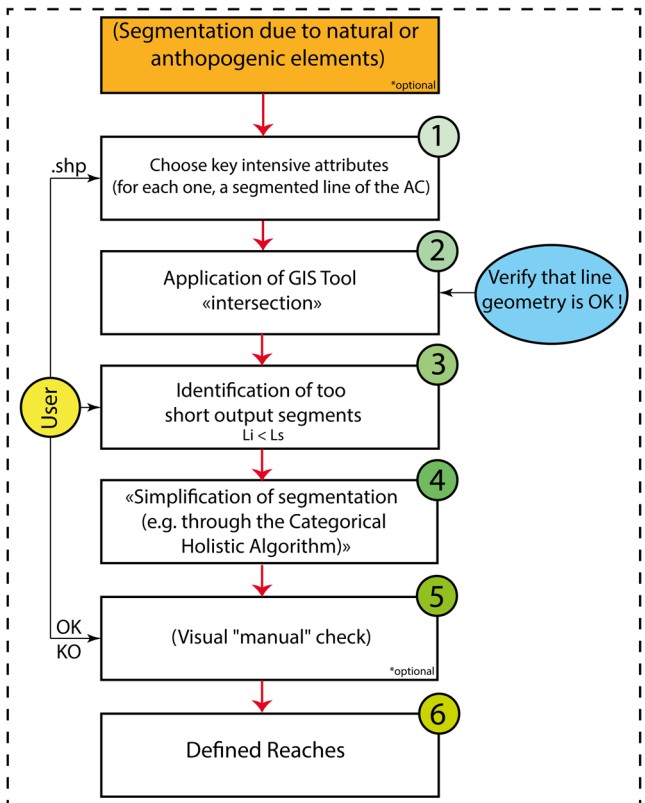

**Figure 4.** The proposed procedure to identify reaches. The necessary User's input consists in choosing the attributes and the significant length Ls; the visual check is optional (and not included within an automatized procedure).

This procedure does not require any particular algorithm (except possibly for the last step) and it does not imply any training nor it is restricted to data sets adopted in the training. In addition, it makes the characters that specify each reach explicit. In brief, a multi-dimensional holistic view problem is solved essentially through a mechanical sequence of mono dimensional operations.

*3.1. Choosing Attributes*

The attributes considered for the segmentation are called 'core attributes'. Many choices are possible, and there is not a fixed prescribed number for them, but the choice can be driven by the following criteria:

- Avoid extensive attributes because they cannot be properly determined without a prior segmentation;
- Avoid attributes describing the "presence–absence" property of geomorphic units (binary attributes);
- Prefer attributes that are clearly recognizable on a map (imagery).

By way of example, relevant attributes are the VB width and the entrenchment (ratio of active channel width over VB width) because they are intensive attributes, and they are visible on a map (or image). Several other attributes can be considered, like active channels' width, number of active channels, bed sediment texture, and bed morphology. Some process attributes, i.e., intrinsically linked to time, like channel activity (recognizable by the presence of bars) can also be adopted; however, in general terms, the characterization of a river indeed just looks at forms as a starting point (including presence of bars, levees, etc.) and only later it infers processes. We do not include hence attributes like "stability" or "activity" (i.e., the rate of movement recorded in the past) because these directly call processes into play. It is relevant to consider that here we are just focusing on reach definition; the investigation of the historical evolution of each reach is certainly useful to fully understand behavior and changes occurred, yet it can come at a later stage (an example at a small-medium scale is provided in Nardini and Pavan, [32] and large scale case study is presented in Chen et al. [33]).

As there often exists a significant correlation between attributes, it is always preferable to select as few attributes as possible in order to avoid redundancy.

For each core attribute, it must be possible to define a proper active channel segmentation where each spatial element assumes a clearly determined value of the indicator measuring the attribute. In other words, core attributes must be spatially discrete. This is clearly an abstraction (e.g., bed material can vary virtually continuously longitudinally—and laterally/vertically), and all the attributes require some kind of spatial averaging, for instance smoothing out where too frequent variations occur along the river. Still, we believe that this procedure is reasonably practical for segmentation and that it can be supported by suitable reductionist-holistic algorithms (e.g., the Hubert test, as proposed by the FCT, for quantitative, continuous attributes, like VB width; or the Categorical Holistic algorithm proposed in Nardini et al. [25,34]).

We consider attributes like bed slope or specific streampower (power of the stream dissipated per unit length and unit width) to be of the extensive type and hence not suitable. This may appear to be a wrong position. Take, for instance, slope; it could be defined virtually in each point, but from a geomorphic point of view the relevant slope is not that one, but rather the average over a still unknown reach. On top of that, if the slope is computed based on the thalweg elevation, severe inconsistencies may occur as the section geometry may vary greatly along any stretch (the slope of the bankfull channel water surface would be preferable, i.e., the VB slope divided by sinuosity, which again depends on the still unknown reach length). In addition, such attributes are not recognizable from a map.

If a scheme like the River Styles Procedural tree is adopted within the RSF, strictly speaking, the core attributes have to be selected among those included in the procedural tree itself because the sought classification is obtained based on such attributes only. Other attributes (like slope, contributing basin area) may be either correlated to the original ones, in which case they are redundant, or not, in which case they are transparent to final classification (and hence useless). This does not mean that information concerning for example location of tributary inputs, or slope changes, should be ignored; this will be considered at the level of reach scale analysis (e.g., the "proformas" envisaged by the RSF) because, with those inputs, a change in flowrate, sediment load, and streampower is typically associated.

Hence, the specific choice of core attributes in relation to the RSF certainly focuses on *planform typology* and *bed material texture* (sediments), as all the others are extensive attributes. The River Styles Procedural tree does not involve VB width directly (actually it does indirectly through *confinement*), and, as such, VB is not essential, strictly speaking, provided that the planform is accurately defined. Anyway, VB width, being a key control factor, may be used in replacement of other attributes that may be missing for some reason, as in the Baker case study considered in Section 4. Notice again that, in principle, there is no pre-determined number of attributes; it is a matter of choice.

At this point, the reader might argue that the problem has remained unsolved as planform typology itself raises the same issues we started from: it is multidimensional as it depends upon

a number of elements, like, for instance, the number of active channels with their size and relative location, sinuosity, spatial pattern of geomorphic units like bars, and so on, requiring a holistic synthesis; and it is not a pure intensive attribute, because several of its characters (sinuosity in particular) depend again upon the segment length considered. Of course, expert-based planform-recognition is again the traditional method (which has also been applied here), but what we are now wondering is whether automation is feasible.

Fortunately, this problem can potentially be solved through AI algorithms as we are just interested in assessing a categorical attribute within a finite set of possibilities. We are not aware of applications to the river context, but there is no conceptual reason why techniques similar to those applied for instance to the case of building spatial pattern recognition (e.g., Li et al. [35]) could not be adopted in the river context. A scale of analysis has to be provided, but a typical criterion of a significant length $L_S$ of about one order of magnitude larger than the width W (i.e., $L_S$ = 10/20 W) should work, where W is the width of the envelope of completely connected active channels. This does not mean that segments of a given attribute SHP polyline all have a $L_S$ length; but rather that differences amongst contiguous segments are searched for with a detail of that order (neither finer, nor coarser). Hence, segments longer than $L_S$ can result, while shorter ones cannot. Additionally, other completely different approaches to automated planform identification are available, as shown in Nardini and Brierley [34]. Hence, we can proceed by assuming that the automated planform recognition is feasible.

### 3.2. Identifying Reaches

To identify geomorphic reaches, we look for a kind of "least common denominator", that is, a suitable segmentation is performed, by proceeding as follows:

- intersection of the segmentations of each "core attribute" thus obtaining potential reaches;
- refinement of the output: too short reaches (with respect to the $L_S$) are eliminated by incorporating them within the closest, major segment, either manually or through a computerized, possibly automated algorithm like the one presented in Nardini et al. [25].

This can all be preceded by considering exogenous cuts in correspondence with natural singularities (e.g., waterfalls, rocky outcrops, isolated constrictions, or tributary inputs) or of anthropogenic interventions (transversal infrastructures like dams or weirs), an objective information generally available (yellow box in Figure 4).

One may say that the proposed method would perhaps not differentiate two consecutive potential reaches which present the same value of core attributes (e.g., sediments and planform), but differ by other attributes, like confinement or sinuosity. According to our logic, they are indeed a unique reach (with a local anomaly), so it is correct that extensive attributes (like confinement and sinuosity, etc.) are computed over its full length. Anyway, should objectivity and automation not be a concern, it is always possible to intervene manually and cut even further the potential reach.

Notice that this procedure, when based on information related to different time instants, is likely to produce different reach results because some of the attributes may have suffered changes. The identification of reaches whose structural lengths (and hence position in space) vary in time indeed makes the analysis of time changes quite complicated. However, this same difficulty also arises with the traditional expert-based approach. More importantly, this whole exercise, by its own structure and contents, can be automated through computer aided tools and procedures.

### 3.3. Avoiding Inconsistencies

Possible inconsistencies may occur if one includes extensive attributes within the core set. Consider, for example, sinuosity: one of the new shorter segments (reaches) obtained by intersection of its segmentation with that from other attributes may possibly present a different ratio of the river axis over the VB axis lengths. From a logical point of view, this is not a mistake, yet it clearly is an undesirable output. The proposed procedure avoids it, as it does not include extensive attributes.

A similar problem arises when a much simpler approach is adopted, which may seem feasible at first sight: to assess all relevant attributes on a regular, exogenously imposed discretization of the river into stretches displaying, say, a significant length ($L_S$ = 10 W), rather than to search for an intrinsically meaningful segmentation first. Such a procedure would lead to inconsistencies because reaches which are physically characterized by a certain value of the intensive attributes—like planform or bed material—would be cut into parts, so all extensive attributes (in particular confinement and sinuosity) would then be computed over segments that mix such continuous attributes. This way, a meaningless output is obtained (Figure 5).

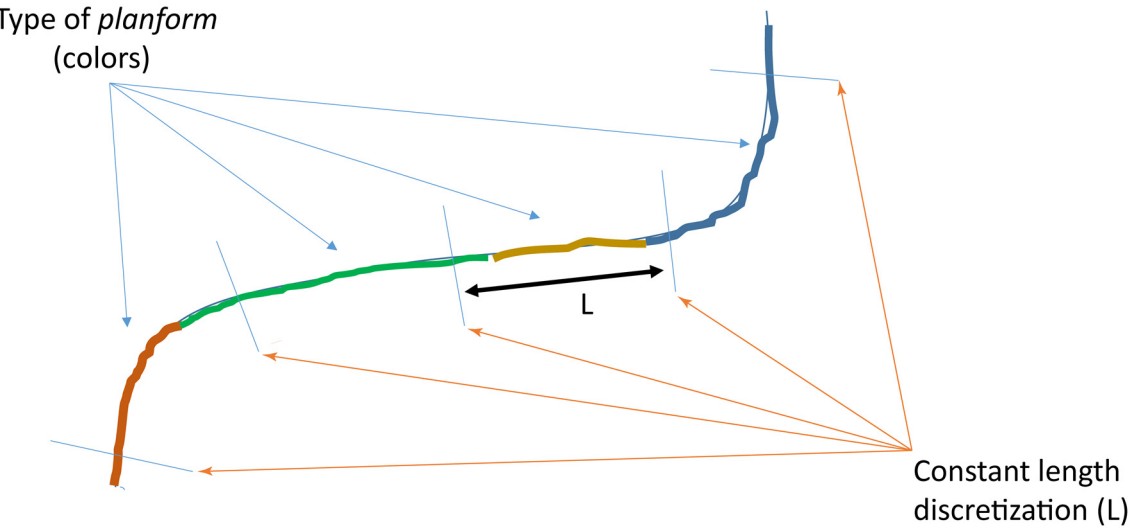

**Figure 5.** Hypothetical exemplification of why a regular, exogenously imposed discretization is not an appropriate solution.

## 4. Application of the Method to the Magdalena and Baker Rivers

In this section, we present the results of a test we carried out by applying our approach to the Magdalena River in Colombia (see Section 4.1) and to the Baker River in Chile (see Section 4.2). We selected these two cases mainly because of two reasons: (a) they are profoundly different particularly because the Magdalena is mostly a low slope, multichannel river (from island braided to anabranching and even anastomosing), while the Baker is prevailingly a medium slope, transitional river (actually, it displays single channel river to wandering and alternate bars typologies, with only few braiding stretches); (b) we could use readily available information suited for the analysis.

A straightforward way to assess the appropriateness of our procedure would be to compare the final segmentation we obtained with that achieved through the traditional, expert-based procedure (see Section 2.1). However, the Magdalena is a long river for which there is no specific segmentation so far, so we decided to proceed differently. After segmenting each stretch with the proposed algorithm, we investigated whether the traditional methodology (where more attributes like Valley Bottom width, contributing basin area, and slopes can be manually included) would produce new cuts that hereafter we denote "novelty". If there are no significant novelties compared to our procedure, the test can be considered successful.

This test, though, is weak because, in order to judge whether a new reach break is to be introduced, basically we just look at the planform type which is one of the core attributes already adopted in our proposed procedure to identify reaches (besides considering the occurrence of visible instream geomorphic units like bars, islands). The test hence focuses on identifying those changes missed in the previous characterization or the possible general inconsistency between the different attributes. This is more meaningful when thinking of the RSF where reaches are identified, characterized, and classified essentially by looking at attributes which are recognizable from a map.

To strengthen the analysis, we adopted in parallel an additional criterion that considers the partly confined stretches and discerns—based on expert judgment—whether they would deserve a further subdivision based on the degree of confinement (one of the traditional criteria adopted); if not, the test concludes satisfactorily. Notice that this subjective information is only used for testing our approach which, on the contrary, only involves objective information.

Of course, here, like in the traditional approach, it is possible to introduce additional cuts according to natural or anthropogenic anomalies. However, as this would work the same way in both approaches, we do not consider it explicitly (and in any case it can be easily automated).

The results of our test are discussed in Sections 4.1 and 4.2.

The Magdalena (Figure 6a) is a large river with an average flow rate of 7200 m$^3$/s (Restrepo, [36]). Its VB axis is about 1400 km long, while the river itself is slightly longer. A segmentation exercise with this magnitude requires time-consuming elaborations and may lead to identifying many diverse typologies (Nardini et al. [37]). In the original study by Nardini et al. [37], the core attributes are planform type, bed material texture, and stream surface (as a substitute of bed morphology, as this is invisible in the murky waters of the Magdalena). It must be reminded that these attributes have been estimated at a preliminary, rough level, as available information is overall scarce and incomplete (see the original GeoMagda project report for details: Nardini et al. [38]).

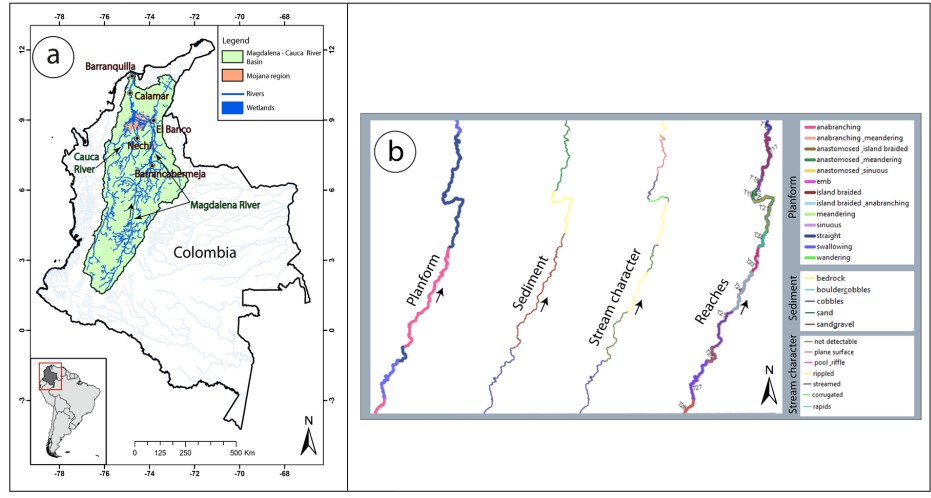

**Figure 6.** (**a**) location of the Magdalena River basin (Colombia); (**b**) Magdalena case study (approximate scale: 1:3,000,000; North on top): definition of reaches for a stretch of the Magdalena River (Colombia) obtained within the GeoMagda TNC-CREACUA project via our method. From the left: segmentation of the planform type, bed material texture, stream surface ("current"). Last on the right: reaches obtained. It can be noticed that, after applying the categorical reductionist-holistic algorithm (Nardini et al. [25]), a short reach of current segmentation (exactly on the elbow) has been incorporated within the T20 reach because it is shorter than the significant length (5 km).

The results obtained by applying our procedure are shown in Figure 6b. In total, 43 reaches were identified.

The Baker (Figure 7) drains a bi-national watershed located in the Chilean and Argentinean Patagonia between 46° and 48° S. The basin has an extension of ca. 28,000 km$^2$, i.e., approximately one-tenth of the Magdalena. It is flanked to the West by the North Patagonian Icefield (NPI). The upper part of the Baker basin (ca. 15,000 km$^2$, 52% of the entire basin area) drains to the Lake Buenos Aires—General Carrera (the second largest lake of South America) and Lake Bertrand (210 m.a.s.l.). The length of the Baker River properly said, i.e., from the General Carrera Lake to its outlet, is about 180 km. It has the highest average discharge of all Chilean rivers: Qav $\cong$ 1100 m$^3$/s (Dussaillant et al. [39]).

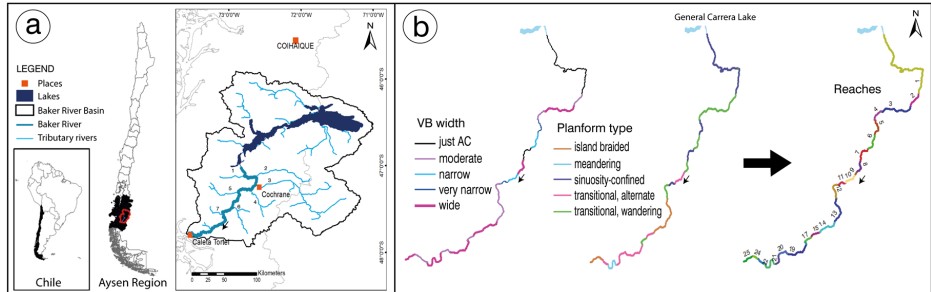

**Figure 7.** (**a**) the basin of the Baker River between Argentina and Chile. Numbers on the right identify tributaries; (**b**) Baker case study: Definition of reaches for a stretch of the Baker River (Chile) obtained with the method proposed here. From the left, segmentation of the VB width; planform type, and the reaches obtained (scale: 1:1,000,000 approx.).

In this case, we adopted the VB width and the planform type as core attributes because no bed material texture data were available. More precisely, a Wolman sediment sampling had been carried out, but it just concerned bars; moreover, despite the notable effort spent, it referred only to a limited number of bars; therefore, it could not provide a sufficiently continuous measure. It must be noted that such data do not show a consistent spatial pattern most probably owing to the considerable, irregular sediment input provided by short, yet aggressive tributaries all occasionally affected by Glacial Outburst Flow phenomena (GLOF, see Dusaillant et al. [39]), which may heavily alter a somehow regular sediment pattern. The real usefulness of such data is evident indeed when studying the GLOF dynamics. As regards planform typology, we defined a stretch as "sinuosity-confined" whenever the curvilinear line linking flexus of the active channel envelope axis lies somewhere outside the valley bottom (VB), i.e., the channel follows the valley bottom axis of a sinuous valley setting with absent or isolated floodplain pockets (sensu Fryirs and Brierley, [40]).

Results are shown in Figure 7b. A total of 25 reaches were obtained.

### 4.1. Case Study A: The Magdalena River

In this section, we illustrate the test procedure already explained, as applied to the Magdalena case. The first criterion, based on slope changes, could not be applied because, except for the upper, mountain stretch, quite marginal, the resolution of the adopted DEM (SRTM 30 m) was too rough. VB width shows several minor changes, a few ones are relevant as shown schematically in Figure 8.

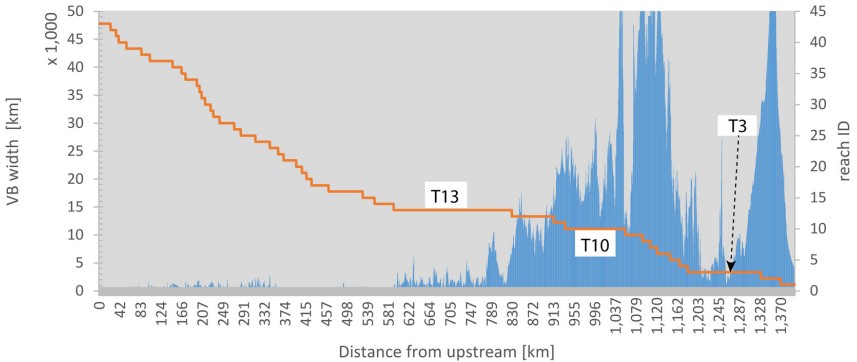

**Figure 8.** Magdalena case study. Schematic representation of VB width (blue line) along the Magdalena (upstream is on the left) showing different reaches determined with the proposed procedure (orange line, with axis on the right). They range from the first one (T43) to the last one (T1) (their vertical position has no relationship with elevation). Reaches T3, T10, and T13 commented about in the following are located here. VB width values greater than 50 km have been cut (actual max is around 80 km) as they do not change the general understanding, while hiding useful details.

In Figure 9a, T13 is the first reach where the VB begins to be significantly present; from this point, visually, it would seem advisable to introduce a further subdivision at its right end because of VB variations. Details shown in Figure 9a demonstrate that the planform in general maintains its island braided character, with just subtle variations (larger and more frequent islands with more tiny side channels in inset B than in A). A more marked difference is that, in A, the river is confined, while, in B, it is only partly confined and the scale would perhaps support this distinction as the significant length $L_s = 10/20\ W$ (W being the width of the active channel envelope) is here about 20/40 km. Finally, inset C shows how the planform, in the short, narrow stretch in the middle, is mostly a straight, single channel; its length, however, is quite short and as such it should be considered a local constriction.

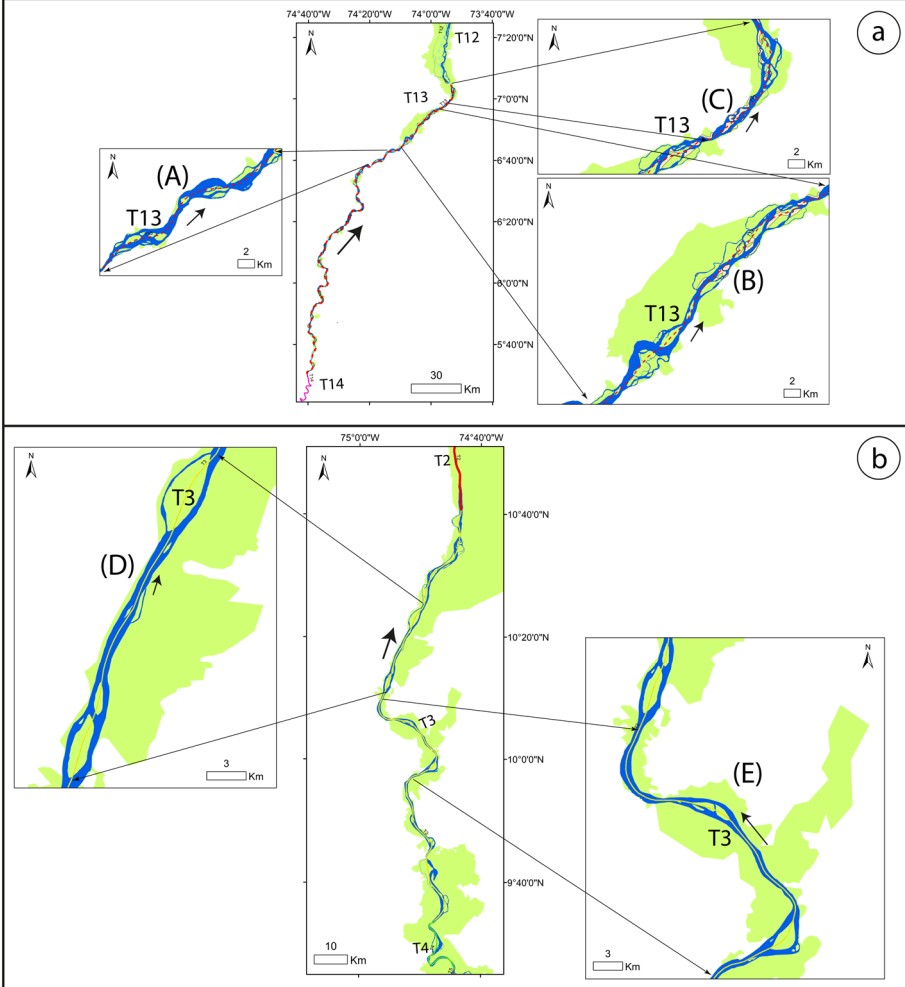

**Figure 9.** (**a**) Magdalena case study. T13 is an example of reach where the VB width criterion might call for further cuts (main stem scale 1:1,000,000; zoom's scale 1:200,000; North on top); (**b**) Magdalena case study. Reach T3, an irrelevant anomaly linked to low DEM resolution in the VB determination (North on top).

As a conclusion, this case shows that the VB width is a key attribute to discern different reaches, but also how relevant the 'significant scale' issue is. In this case, by taking position on the lower bound of the significant length, additional reaches (based on the planform criterion itself) could be identified, while this would not be the case if the upper bound was chosen.

Another emblematic case, according again to Figure 9b, is represented by reach T3 that seems unaffected by a couple of VBs narrowing locations and a central peak. As visible in Figure 9b, the narrowing (or the enlargement) in fact does not change at all the planform character of T3 (compare

insets D and E); actually, the anomalous, local VB widening has to be attributed to an imprecision in the original DEM (SRTM 30 m) adopted for its determination by the Fluvial Corridor Toolbox.

Figure 10 illustrates the analysis of main tributary inputs (remind that we use here "novelty" to denote a new cut of a stretch):

(a)   Rio Suaza (reach T38, T39): no novelties are introduced with respect to the results obtained. There is an almost coinciding reach break due to the attribute "current", which switches from "corrugated" to "pool & riffle", yet, considering the low precision of that attribute, it does not introduce a true novelty.

(b)   Rio Paez (reach T37): apparently a novelty is introduced as the AC width (light blue) slightly increases, but, in reality, this effect is due to the nearby downstream Betania Reservoir that induces a backwater effect which was neglected in the original analysis (to keep consistency with the rest of the available information).

(c)   Rio Cabrera (reach T26, T25): no novelties are introduced locally. A planform change was already detected downstream (upwards in the figure) switching from reach T26 to reach T25 owing to a change in the planform (from single, straight to anabranching) as well as in the sediments (from cobbles to sand & gravel). That switch may be due to the sediment load of the tributary; however, there is a local increase of VB width which probably is the main controlling factor.

(d)   Rio Prado (reach T24): The seemingly local, downstream widening occurs in several locations along this anabranching, sand and gravel, rippled surface reach.

(e)   Rio Saldaña (reach T23, T22): a planform change was already identified (from anabranching to straight, sinuosity-constrained), mainly due to the VB width change. A confluence bar and some downstream bank attached bars appear, yet 6 km upstream there already was one. Therefore, here again, no novelties are introduced.

(f)   Rio Sumapaz and Bogotá (reach T21): a change of the attribute "current" had already been detected though it was almost certainly due to a switch of the satellite image observed. The presence of an island right before the Bogota River input may be linked in some measure to the sediment input and backwater effect induced by the two tributaries, as well as to a slight local slope change (not detectable with the information at hand). In any case, essentially the tributary inputs do not introduce novelties in this sinuosity-confined, bedrock stretch.

(g)   Rio Negro (reach T13): this important tributary does not introduce significant novelties in the planform, or in the VB width or the presence of instream units (bars, islands) which characterize this island braided, sand, rippled reach.

(h)   Rio Carare and Rio Opón (reach T13): no novelties are introduced by these two rivers.

(i)   Rio Sogamoso and Rio Cimitarra (reach T13, T12): there is a change of planform with an evident anabranching character from right before the important Sogamoso River input; this, however, was already detected, so no novelties are introduced.

(j)   Rio Lebrija (reach T11, T10): no novelties are introduced by this relatively small tributary, as a planform change had already been detected from a still anabranching character in T11 to a prevailing anastomosed character in Reach T10.

(k)   Rio Cesar, Cauca, and S. Jorge (reach T10, 9, 8, 7, 6, 5, 4): the influence of the important Cesar River is significantly moderated by the large wetland (the "Ciénaga La Zapatosa") at its outlet into the Magdalena River. This wetland, analogously to many other wetlands in the area, regulates the river–wetland water exchanges in both senses with a strong effect on the dynamics of the fish population (Granado-Lorencio et al. [41]; López-Casas S. et al. [42]). In the whole area, the  Magdalena maintains its anastomosing character until the exit from the Mojana region (reach T4), with several character variations of the main stem (see "brazo La Loba") and additional branches. The only apparent effect of these tributaries seems to be a meandering pattern of the Magdalena right before its confluence with the Cauca (the most important one in the basin).

These tributaries, particularly the Cauca, are of key importance in terms of water flows and hydrological regime; however, no novelties in the segmentation can be detected.

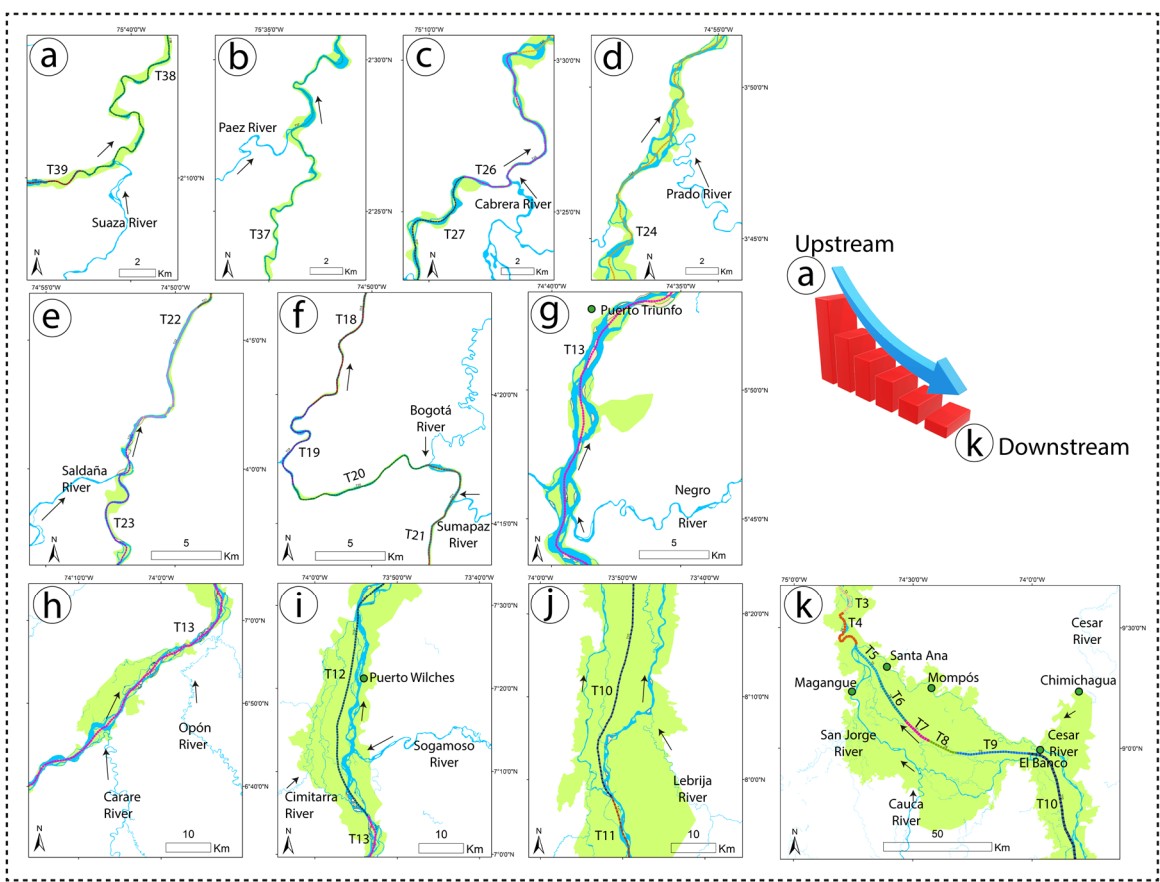

**Figure 10.** Magdalena case study. Stretches (from upstream, top left, to downstream, bottom right) where a significant change of contributing basin area occurs, i.e., a tributary comes in (figure k scale 1:3,200,000 approx.; all other zooms scale 1:500,000 approx.). The figure also shows active channels (blue), VB (light green), islands (dark green), and main bars (dotted yellowish) according to official IGAC information. Colors and numbering (Txy) along the river identify the reaches.

In conclusion, we can reasonably state that the segmentation obtained is not modified by the tributaries input (see Figure 10).

Let us now consider the partly confined stretches (Table 1) to assess whether these should have been further split. According to the characterization described in (Nardini et al. [38] and Nardini et al. [37]), conclusions are as follows:

**Table 1.** Magdalena case study: partly confined reaches of the Magdalena river according to the River Styles analysis performed on the segmentation obtained with the proposed procedure in Nardini et al. [37,38].

| 1 | 2 | 3 | 4 | 5 | 6 | 7 | 8 | 9 | 10 | 11 | 12 | 13 | 14 | 15 | 16 | 17 | 18 | 19 | 20 | 21 |
|---|---|---|---|---|---|---|---|---|----|----|----|----|----|----|----|----|----|----|----|----|
| T1 | T3 | T4 | T6 | T13 | T23 | T24 | T25 | T27 | T28 | T29 | T30 | T32 | T33 | T34 | T35 | T37 | T38 | T39 | T40 | T43 |

Starting from the upstream reach T43, Figure 11a shows the pattern of the active channel within the VB and the type of confinement (visible from the position of the active channel within the VB corridor). It is quite evident that no further subdivision would bring in additional light regarding its character: the whole reach is a partly confined, sinuous one. This situation is emblematic of several other reaches (T40, 39, 38, 37, 35).

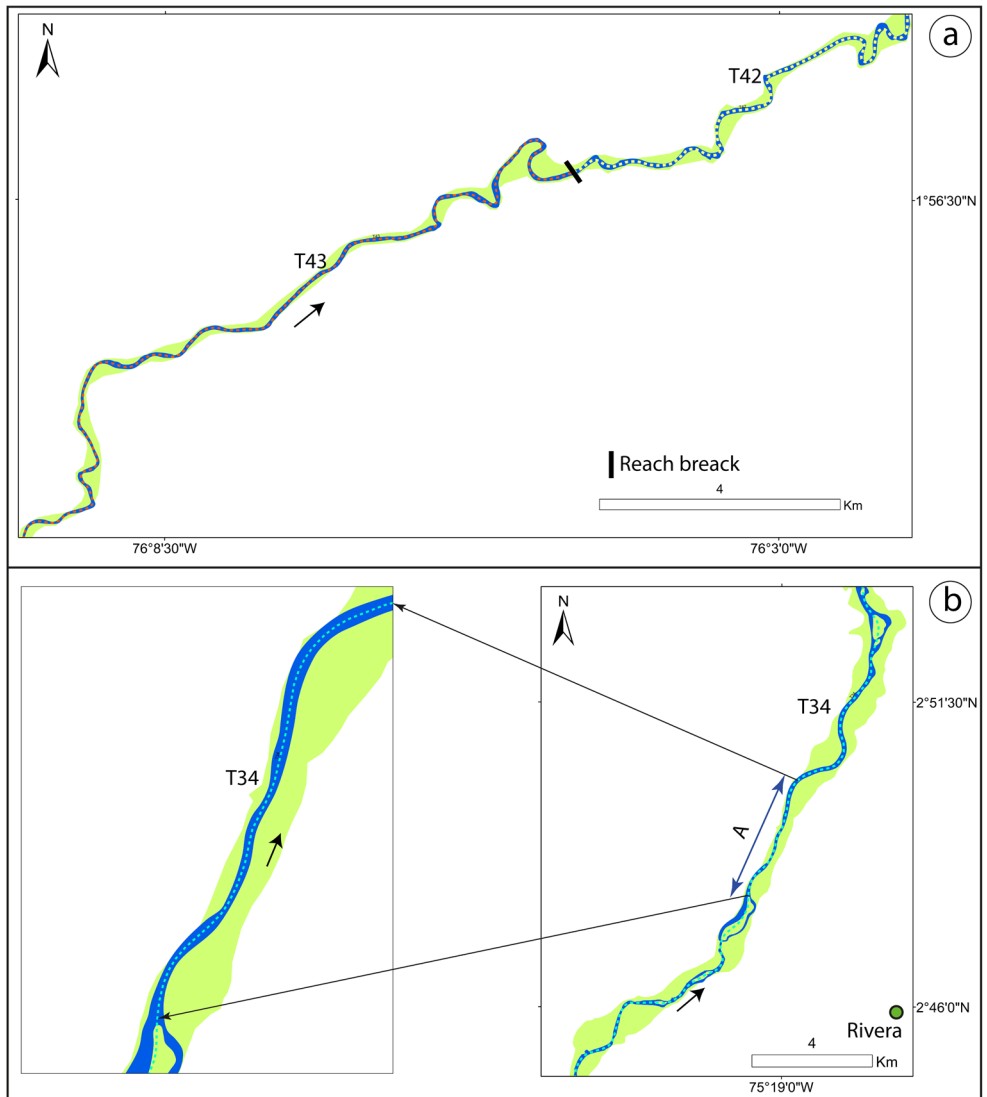

**Figure 11.** (**a**) Magdalena case study. Character of an emblematic partly confined reach (T42–T43): Any further break would not allow significant changes to the river character (the break between reaches T43 and T42 is due to the attribute "current"); (**b**) Magdalena case study. Example of ambiguity: partly confined Reach T34 where sub-reach A appears to be confined, but it is not (North on top).

Reach T34 seems analogous, yet it deserves a further insight. As apparent from Figure 11b, taken as a whole, the reach is a partly confined one; still, if a sub-reach, like A, is considered, it could be classified as "confined" as most of it appears to be bounded by the left VB margin (look at the complete figure on the right). A similar situation holds for other sub-reaches. However, this is not a correct conclusion. Indeed, when carefully analyzed, even reach A reveals to be not fully in contact with its VB margin, as shown in the figure inset; moreover, the assessment depends on the chosen spatial scale. According to the general criterion earlier proposed of a significant reach length $L_s$ = 10/20 W, in this case, one gets $L_s$ = 5–10 km: therefore, the initial assessment (partly confined) is confirmed. Thus, no novelties are introduced with respect to the original segmentation. The same observation holds for all the other partly confined reaches (T13 was already commented in the VB section).

In conclusion, the segmentation of the Magdalena River obtained with the proposed procedure looks reasonably accurate and robust. The VB width appears to be a key factor and, as such, it should be in general adopted amongst the core attributes, while the attribute "current" seems rather inadequate (it was adopted as a trial hypothesis in the cited original study).

### 4.2. Case Study B: The Baker River

In this section, we illustrate the test procedure already explained, applied now to the Baker River. Ulloa et al. [43] presented a segmentation of 34 reaches. However, it is not appropriate for the purpose of comparison. In fact, the procedure adopted there was not exactly the "traditional" one, but rather a 'mixed' one: reaches were defined by expert judgment, mainly based as usual on a qualitative appreciation of confinement; then slope, braiding index, and sinuosity indices were computed and considered. However, quantitative indices, as noted before, while being objective measures, are intrinsically linked to the initial, arbitrary segmentation choice (they have an extensive nature); furthermore, they are strongly affected by measurement details (e.g., precision of altimetry, choice of sites where elevation is assessed; how many transects are considered to compute the braiding index). Therefore, the numerical output of indices inevitably introduces reasons for additional differentiation of reaches which carry little geomorphic information, particularly when a RSF is considered.

For this reason, we did not use the Ulloa et al. [43]'s segmentation for comparison purposes; instead, we proceeded as in the previous case study, by investigating whether some further reaches ("novelties") could be identified with respect to those generated by our proposed procedure once additional attributes were considered (Figure 12).

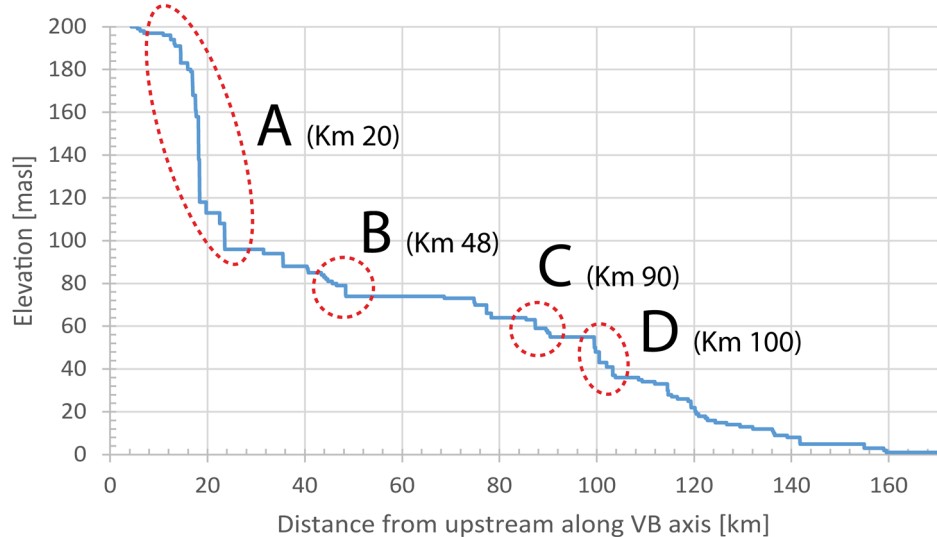

**Figure 12.** Baker case study: approximate elevation of the VB along the river axis (from USGS Earth Explorer, Digital Elevation: GMTED2010, Entity ID: GMTED2010S50W090, 11 November 2010).

We start now with slope changes (considering the elevation of the VB along the axis of the active channel envelope).

As it is apparent from Figure 12, the sharpest elevation drop (and presumably slope change) occurs at location A, just before km 20 and once more just after it. The inset in Figure 13 shows that indeed the river presents an anomaly there with two isolated very large bars; right downstream, the two falls. Although worthy of attention, this is, however, not suitable to introduce a novelty, being in fact just a local anomaly. The jump at km 36 does not reflect in any of the local visible planform and geomorphic units.

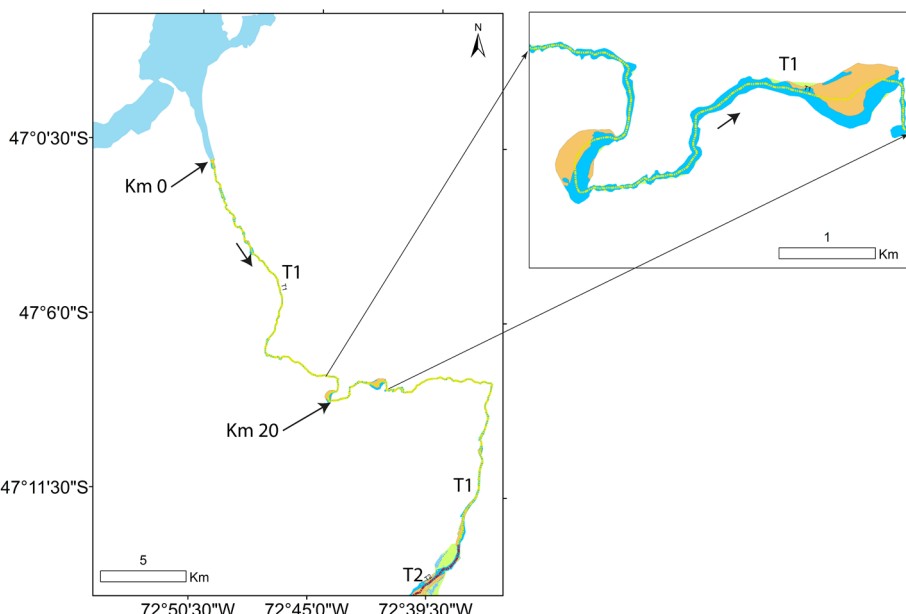

**Figure 13.** Baker case study: detail of the river at the major slope change (location A in the elevation profile).

Location B at km 48 (Figure 12) incidentally coincides with the point where a change of planform was already established between reach T3 and T4 (see Figure 14, bottom left) and hence does not introduce any novelty.

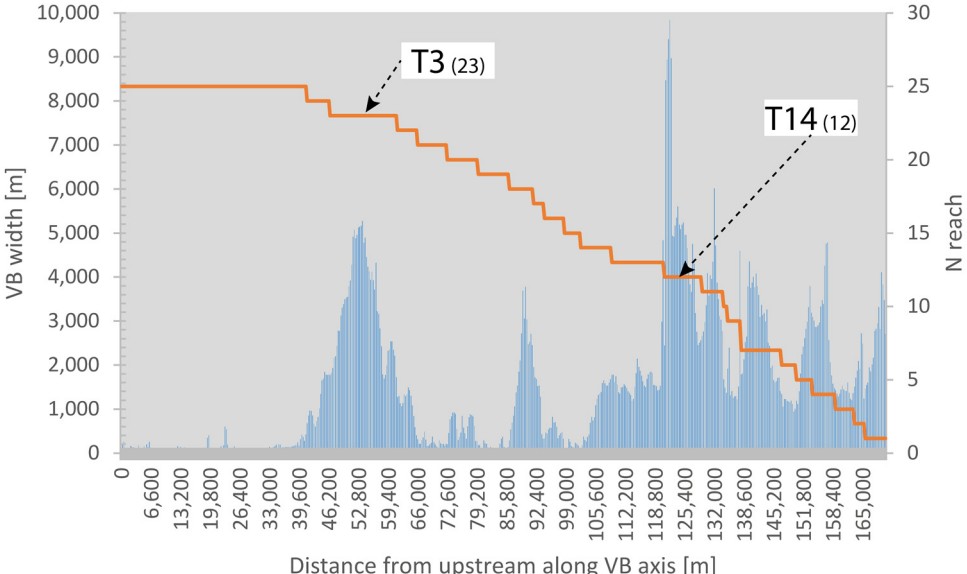

**Figure 14.** Baker case study: schematic representation of VB width (blue line) along the Baker River (upstream is on the left) and of the different reaches determined with the proposed procedure (orange line, with axis on the right); they range from the first on the left (T1, denoted as 25 on the right scale) to the last one on the right (T25). To improve visual perception, VB width values greater than 8 km have been cut (max is around 10 km) as they do not alter the general understanding, while hiding useful details (note: the numbering of reaches on the right axis does not relate to elevation; it is the complement of the one adopted in the following figures, for ease of representation here).

This example is emblematic for several other jumps of similar magnitude (e.g., the one at km 87 between reach T7 and T8; and the one at km 100 approx. between reach T11 and T12, at location C and D, respectively, in Figure 12) where the changes are even more evident (see, for instance, Figure 15).

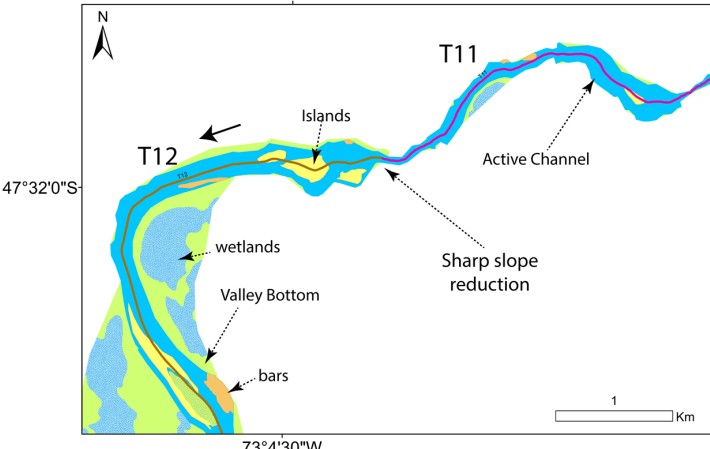

**Figure 15.** Baker case study. Example of typical coincidence of a reach boundary with a slope change and a VB narrowing (may be widening in other situations): end of segment D at km 100 approx. (Figure 12a). The longitudinal arrow indicates flow direction.

Figure 14 shows a few quite relevant changes as well as several minor changes of the VB width (assessed by means of the Fluvial Corridor Toolbox). In this figure, the first reach which may raise some suspicion is T3 (N° 23), where the VB width ranges from less than 2 km up to about 5 km. The details shown in Figure 16a (top left) demonstrate, however, that the general planform character is the same within that reach. A possibly distinct character (alternate bars) might be detected in the short reach of the inset, but its length does not pass the significant length criterion (here about 5/10 km) to be meaningful and would any way not be linked to the VB width.

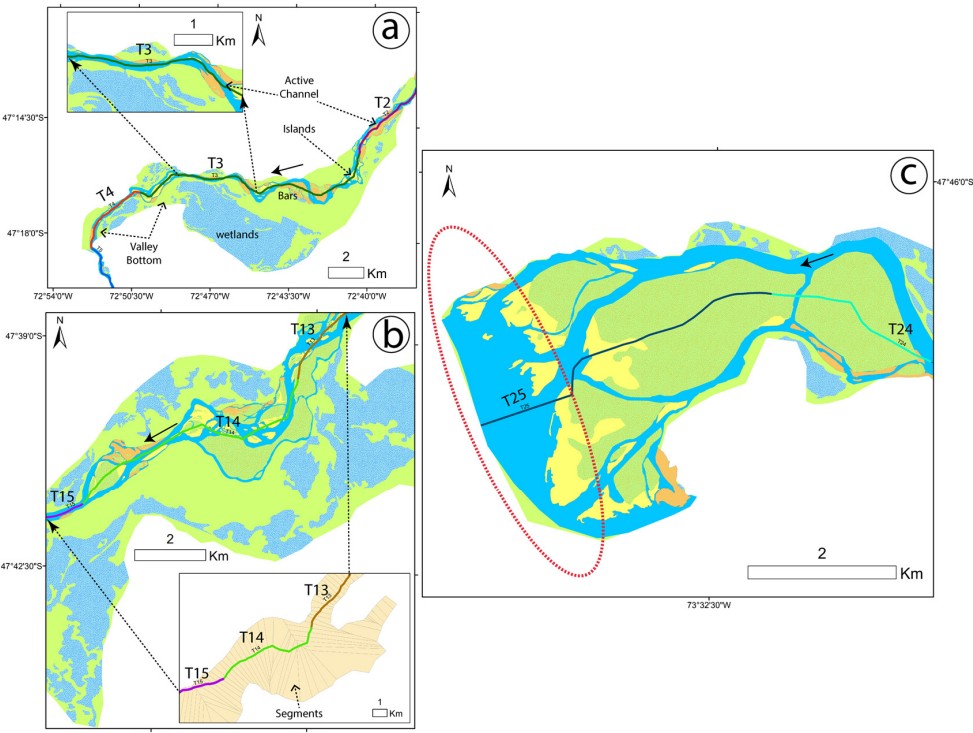

**Figure 16.** Baker case study: (**a**) details related to reach T3 (top left, denoted as 23 in Figure 14); (**b**) T14 (bottom left): no novelties are introduced by the VB width change criterion. Flow direction follows increasing reach numbers and (**c**) Baker case study: the last reach deserves further splitting (North on top).

Perhaps another situation that is worth considering is that of reach T14 (N° 12) where the VB ranges from 2.7 km up to 9.8 km (see Figures 12 and 16b). The details shown in Figure 16b demonstrate again that the VB variations do not support relevant changes in the planform or other characters at the significant scale. Moreover, it must be noted that the VB width values are somehow artificially exaggerated by the FCT algorithm adopted that produces situations like this very irregular polygons, while partitioning the VB. This fact may dramatically change their orientation and, as such, their width; in addition, the kind of queue on the top right seems more likely to be a legacy of an ambiguous cut of the Baker VB from that of a local tributary, rather than a meaningful geomorphic element. Very similar considerations hold for the other situations, except for the last T25 which, as shown in Figure 16c, would have perhaps justified a further subdivision as its last part is clearly a transitional area seemingly very much influenced by tides. This would, however, not be linked to the VB width criterion.

With respect to the tributaries, Figure 17 (starting from upstream) clearly shows that no significant novelties are introduced by tributaries 1 and 2 (perhaps just a slight narrowing of the active channel for tributary 1—associated with a bed fall—and instead a slight widening for the second; both, however, are too subtle to determine a meaningful differentiation).

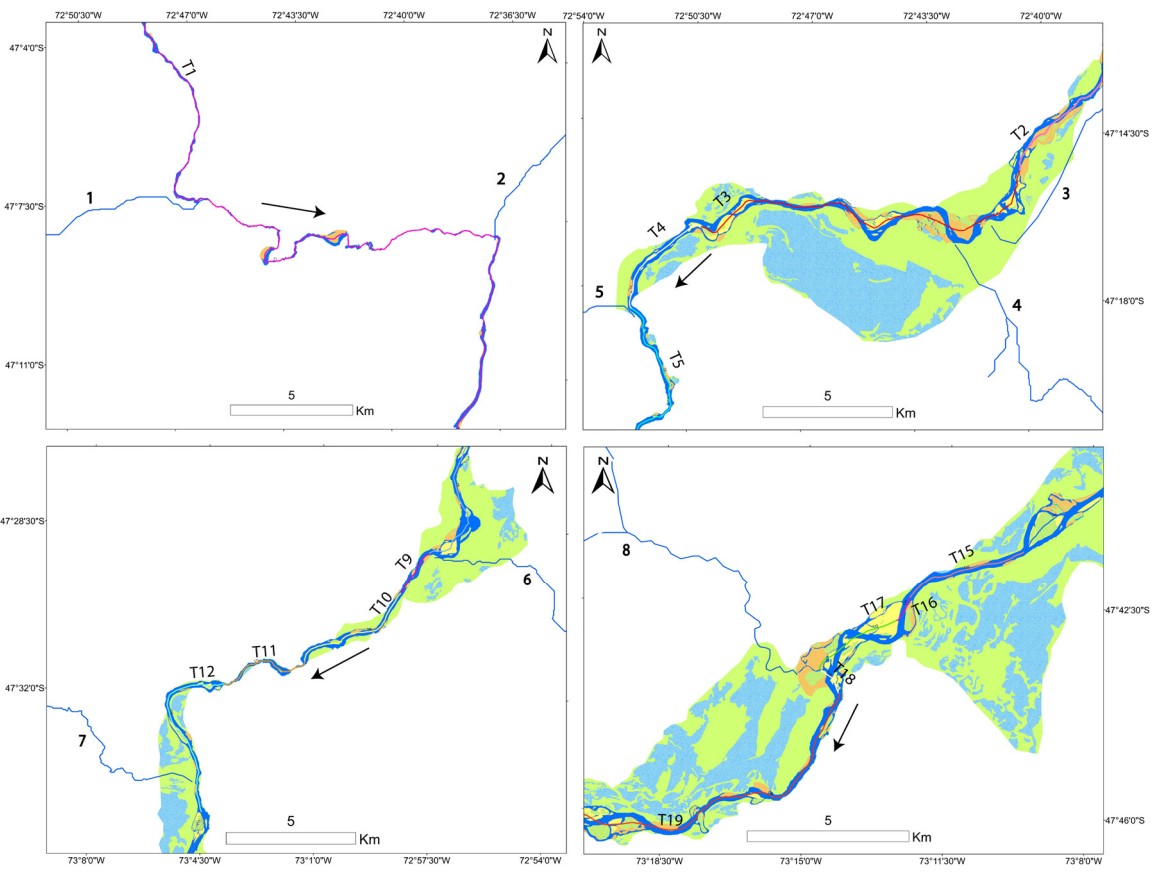

**Figure 17.** Baker case study: investigating the effect of tributaries.

Seemingly, tributaries 3 and 4 are associated with the significant VB widening and change of planform style, already captured by the segmentation. However, this widening is more probably generated by the accumulated material carried by tributary 5. This one—as visible from Google Earth imagery—has a very high sediment load from the western ice field. Tributary 5 demarcates a significant narrowing, already captured by a planform change. Tributary 6 seems to contribute to the generation of large bars right upstream its confluence, a fact already captured by a planform change that subsequently becomes "transitional, alternate bars". Tributary 7 seems instead to be associated to an increment in sediment load that gives rise to a change of pattern of bars and islands, a fact already

captured by a change in planform typology right downstream of its confluence (from a transitional, wandering into an island braided typology). In addition, this last tributary seems to introduce a change that has already been captured by the planform attribute (transitional, wandering until upstream of its confluence and transitional, alternate bars downstream).

Analogously to the Magdalena case, tributaries do not introduce significant novelties to the segmentation based on map view (though they, of course, may change the water flow, sediment load, and streampower).

As for partly confined reaches, the first one from upstream is reach T3 which has already been mentioned above: no novelties are introduced by the confinement criterion as visible in Figure 13. Another partly confined reach worth considering is T12 (Figure 14, bottom left); however, the split of it into confined and not confined sub reaches would clearly violate the significant length criterion while not shedding light on the planform type. Totally analogous considerations hold for the following partly confined reaches (T13, T15, T19, T21, T22, T23).

## 5. Discussion

The test presented in the previous section has performed successfully in both case studies, as we found that the segmentation obtained through the main criteria usually adopted in the traditional approach would not differ from the one achieved with our proposed method. Valley bottom width appears to be a key factor, as already assumed by Bertrand et al. [30] and then proven, for example, by Martinez-Fernandez et al. [26]. Slope changes do not provide additional information to that already conveyed by the chosen attributes. All of this supports the proposed criteria for the choice of core attributes as there is an evident correlation amongst them, particularly as regards changes in VB width, slope, and planform typology. Given that these observations refer specifically to the two case studies considered, further research is needed to assess whether they can hold in more general terms.

In Annex A of the Deliverable D2.1, Part 2 of the REFORM project (Bizzi et al. [4]), the experience presented apparently resembles very much the approach we are proposing here. However, our procedure introduces some important improvements: (i) it explicitly addresses the paradox pointed out in Section 2.1 (reaches → characterization →reaches); (ii) it explores a completely different technique from the statistical, mono-dimensional Pettitt (or Hubert) test, through a new logical-heuristic holistic algorithm; (iii) it clarifies how sporadic and extensive attributes (like presence/absence of geomorphic units and confinement) can be incorporated; (iv) it provides criteria to select core variables in general and specifically for a River Styles oriented analysis; (v) it explicitly exploits the areal segmentation of the VB (obtainable for instance by the Fluvial Corridor Toolbox) to assess presence–absence binary indicators, very suited for dealing with geomorphic units; and, in addition, (vi) it avoids redundancies and the whole procedure to determine reaches is well defined.

Bertrand et al. [30] present a logical-heuristic approach that includes criteria very similar to the ones we propose here. Their aim is to identify reaches with different and meaningful river typologies (because of sediment replenishment interventions). However, they do not address explicitly the core circular problem underlined here (reaches → characterization →reaches); they somehow bypass it by identifying "homogeneous units" based solely on active channel width and VB width. Then, they define a large number (30) of attributes with associated quantitative or qualitative (categorical) indicators, for each "slice" of the discretized river. Some of these attributes are indeed extensive and as such defined and calculated over the pre-defined homogenous units (i.e., segments). A first screening of such attributes is done with the help of the Principal Component Analysis to identify the most significant ones (because independent); then, a classic cluster analysis is performed to identify river typologies with their associated reaches. Strictly speaking, this process leads to a certain degree of inconsistency because the extensive indicators should now be computed over the reaches that are found, not over the original homogeneous units, hence requiring iterations, at least in principle. Our new method directly addresses this point and avoids this inconsistency providing therefore an improvement. Another aspect worth noting is that their attributes are chosen on the basis of criteria

that are rooted in the theoretical understanding of riverine ecosystems and joined with engineered thought, thus leaving to a logical-heuristic algorithm (clustering) the responsibility to reveal which are the attributes that carry indeed more information. However, adherence to the River Styles scheme, as done here, leads to an identification of reaches, and their characterization, which supports a more straightforward behavioral interpretation based on an established logical scheme (the RSF of Brierley and Fryirs, [5]).

The SOM algorithm adopted by Bizzi and Lerner [27] and the MRPP algorithm adopted by Martinez-Fernandez et al. [26] seem quite promising as they use multidimensional information that, in principle, might lead to the sought holistic synthesis of the selected attributes. However, they cannot deal with categorical attributes, while our method can.

## 6. Conclusions

The procedure here described provides a simple and original solution for the segmentation problem for identifying river reaches. The underlying approach looks for a holistic synthesis from reductionist information while focusing on the distinction between intensive and extensive attributes and hence it avoids the "paradox" identified in Section 2.1 in relation to the manual segmentation approach based on expert judgement. This new procedure is very simple, systematic, and objective. It also clarifies how other important sporadic or extensive attributes like confinement or presence/absence of geomorphic units can be incorporated. In addition, the output has double added value with respect to the River Style characterization problem: in fact, reaches are defined and an intrinsic consistency with the River Styles framework is ensured by construction. We believe that, thanks to this, the River Styles approach can become more applicable by practitioners than it currently is.

This paper supports a step forward in the automation of the whole process of river analysis. In the near future, algorithms for automatic satellite image interpretation will enable a prompt and reliable identification of the active channel and its geomorphic units inside the channel and within the floodplain (e.g., Bertrand et al. [30]; and Fryirs et al. [40] for a recent review; and, for instance, the CHAMP project: https://www.champmonitoring.org/). At that point, the whole River Styles characterization can be automated: first, the assessment of intensive attributes (mono or multi attribute reductionist-holistic exercise), then the identification of reaches (as explained in this paper), and finally their classification (see Nardini et al. [25]). Expert judgment is not ignored, of course, as it drives the interpretation of behavior. We openly encourage further investigation into how our approach can perform in other cases.

**Author Contributions:** Conceptualization: A.N. and S.Y.; Methodology: A.N.; Software: S.Y. and H.U.; Validation: B.M., H.U., M.D.B., and A.L.; Formal analysis: A.N. and M.D.B.; Investigation: A.N., S.Y., B.M., M.D.B., H.U., and A.L.; Data curation: A.N., S.Y., B.M., and H.U.; Writing—original draft preparation: A.N.; Writing—review and editing: A.N., S.Y., B.M., H.U., M.D.B. and A.L.; Project administration: A.N. and B.M.; Funding acquisition: A.N. All authors have read and agreed to the published version of the manuscript.

**Funding:** This research received external funding thanks to The Nature Conservancy through the "GeoMagda" Project, contract No. TNC-CREACUA NASCA 00162/2018. Héctor Ulloa's participation was made possible by FONDECYT Project 3180109 "Análisis geomorfológico y de vegetación de ríos afectados por erupciones volcánicas". Bruno Mazzorana's participation was made possible by FONDECYT Project Nr. 1200091 "Unravelling the dynamics and impacts of sediment-laden flows in urban areas in southern Chile as a basis for innovative adaptation (SEDIMPACT)".

**Acknowledgments:** We thank Gary Brierley and Kirstie Fryirs for inspiring this work with the River Styles Framework and fruitful exchange conversations.

**Conflicts of Interest:** The authors declare no conflict of interest.

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
