# Peer review of "A Systematic, Automated Approach for River Segmentation Tested on the Magdalena River (Colombia) and the Baker River (Chile)"

_water, doi:10.3390/w12102827_

Round 1

Reviewer 1 Report

Review Nardini et al. Geomorphic river characterization: a systematic,automated approach for river segmentation tested on the Magdalena River (Colombia) and the Baker River (Chile) As someone who deals with this type of river reach segmentation problem in my research, I can appreciate the difficulties of automation of this process. The authors have correctly identified the problems, including the very fuzzy “definition” of reach. I believe that the manuscript merits publication after minor revision (see my notes below). The English usage can be improved and I have made some suggestions for that. Please also note the places where your discussion rambles a bit (especially between lines 450 and 664) and where it can be tightened up to improve readability. Rethink your conclusions and make a stronger argument as to why this approach is an important improvement. Recommendation; Accept with minor revisions Suggested changes Abstract: 21 This paper proposes a systematic procedure to identify river reaches from a geomorphic 22 point of view. Essentially, we aim to obtain a holistic synthesis from a reductionist, multi- 23 dimensional information basis structured according to a discretization of the axis of the river. Our 24 approach virtually does not require expert judgment once criteria are set and, as such, it can be 25 automated. The sentence starting on line 22 at “Essentially… of the axis of the river” is a bit jargonistic. I would suggest rewriting that sentence in plain English so a reader who is casually looking at abstracts will have a better idea of what the authors have achieved here. The abstract in general should be expanded to show some real results! Lines 45-46 are too short of a paragraph. Just remove the new paragraph at line 47. Figure 1 is a nice and concise definition of the problem! In the caption on lines 97 - 100 those last sentences could use some rethought. Possibly change from; “In addition, when the river moves (just adjusting without changing its character), “confinement” may possibly change; as it is the starting point of River Styles procedural tree, this may imply dramatic changes in the classification and hence it may require a lot of work to be re done.” To something like; “In addition, when the channel migrates (adjustment without changing its character), “confinement” may possibly change; since this is the starting point of River Styles procedural tree, this may produce dramatic changes in the classification and hence significant resegmentation work.” Lines 106 and 107 “powerful, as shown for instance by Jacquez, [15]), being able to interpret the whole assemblage of a wide array of elements like spatial configuration, altimetry, “colors” through image comparison. Needs to be expanded. Most readers will not know what you mean by “colors”. You use it again in figure 4 without much in the way of explanation. So it needs to be better defined here. Also, “the whole assemblage of a wide array of elements…” can be written as “an assemblage of a wide array of elements…” Lines 108-120. While I agree with much of this our own research using neural networks to identify curvilinear features on Mars has shown that the newer CNN approaches, while time consuming, can indeed get around some of these problems. I suggest that the authors look at RetinaNet as a possibility. I’m not suggesting a major change to this paragraph, but perhaps tone down the negativity towards this approach since some of the major objections they raise can be read as a false “straw man” argument- designed to be easily knocked down. Lines 144-146 144 Hubert [22] introduced a statistical test to define reaches which works in a similar manner; it is 145 currently considered the key 'tool' to support reaches identification within the Fluvial Corridor 146 Toolkit (Roux et al. [19]). Yet, the output of Hubert’s test can be questionable (Figure 3). The real important information of the questionable results are buried in the caption for Figure 3. I would suggest moving the following text from lines 158-161 in the caption “the output (stretches of different colors) is questionable. For instance, the stretch marked by the two red segments would certainly be classified by expert judgment as narrower with respect to the one above it (notice that there is no criterion linked to the length as Hubert test determined both short and long segments) (source: Nardini et al. [24]).” to the end of line 146 and rewriting the short 144-146 paragraph for clarity. Line 198 and afterwards. The authors use “River Styles Framework” repeatedly after this point in the manuscript. Why not just say in line 198; “198 pervades the River Styles Framework (hereafter denoted RSF) (Brierley and Fryirs [5]).” Lines 227-233. An excellent point! Might need to emphasize this a bit more. Lines 260-277. A bit of a long paragraph. I would suggest the following; “…often exists a significant correlation between attributes. Therefore, it is always preferable to select as few attributes as possible in order to avoid redundancy.” Then new paragraph starting at (and with suggested minor wording change): “266 For each core attribute it must be possible to define proper active channel segmentation where each spatial element assumes a clearly determined value of the indicator measuring the attribute. In other words, core attributes must be spatially discrete.” I note that this is an important point and is clearly a weakness in many of the other approaches to segmentation. Line 278 What do you mean by “specific streampower”? and should it be stream power rather than streampower? Line 285 again and like my previous comment; “285 If a scheme like the River Styles Procedural tree is adopted” add (hereafter RSP) to avoid multiple times spelling this out further in the text. Lines 299-318 are close to a stream of consciousness type argument. I would rethink and rewrite this paragraph as two paragraphs with the split coming at line 306 at; “Fortunately, this problem can…” Line 328 at the start of this short paragraph is awkward; 328 All this can be preceded by considering exogenous cuts in correspondence of natural 329 singularities (e.g. waterfalls, rocky outcrops, isolated constrictions, or tributary inputs) or of 330 anthropogenic interventions (transversal infrastructures like dams or weirs). and should be rewritten as “This can all be…” Line 385 You somewhat counter your own argument by inserting on line 385 “-based on expert judgement-“ I think your readers will, like I did, think “Wait. Isn’t this supposed to be an automated approach?” So I would further explain what “expert judgement” is here and why it does not detract from your approach and stated goals. I note that my PDF is missing Figure 5 and 6 and I hope that the authors have done a good job on these important reference figures. On line 450 you first use the term “novelties”. You then use the term 16 more times in the manuscript but never define what a novelty is. It would be easy to define it in line 450 so the reader (including me) might actually know what a “novelty” is! In lines 450 and following through to the beginning of the discussion on line 664 the writing is a bit confusing and should be rethought and rewritten for clarity. It is a longish section and an important one yet it fails to make the case as strongly as it could. In other words, I think the authors have a good approach but could make their case more strongly here and in the Discussion and Conclusions sections.

Reviewer 2 Report

This manuscript identifies some river segmentation approaches and describes another procedure followed by applications in two case studies.

This topic is worthy of research. In fact, properly longitudinal geomorphic segmentation and characterization of the main channel and its valley is of utmost importance. An example of this is the river segmentation for flood monitoring. Also for numerical modeling purposes, river flow models that consider satellite observations of the water surface elevation require meaningful segmentation into reaches.

However, after reading most of the references that appear in the manuscript, and other publications on the same topic, I cannot identify in this manuscript anything substantially new and of scientific value for the international scientific community.

In several parts, this manuscript looks like a technical report, including appendices (just an example: Figure A5 is mentioned instead of Figure 9). In addition, two Figures are not provided. Essentially, it just adds another application of the same methodology already published by the same authors.

Indeed, this manuscript follows a (public) report "MANUAL Toolbox GeoMagda" and (at least) two recent publications (2020) (in Geosciences and Water journals) by the same authors on the same topic. So it is, essentially, more of the same.

What really matters is, in fact, to distinguish what is state-of-art (in scientific terms) and what is a novelty (if any). Indeed, there seems to be nothing significantly new and with scientific value to justify another article on the same topic.

A long and descriptive manuscript, largely based in the bibliography, with numerous 'clichés' or vague meaning sentences and a number of trivial concepts, does not constitute sufficient grounds for a scientific paper.

Following previous publications by the authors, among others, I may suggest resubmission of the manuscript as a “Case Study”, after a thorough reformulation, providing and discussing possible comparisons with different procedures and objectives.

It must be kept in mind that the attributes and key factors must be valued according to the purpose of the river segmentation.  

Reviewer 3 Report

The manuscript is really difficult to read. Sometime the points are quite obscure. In addition, it looks like the authors prefer to use really long sentences with two or more subordinate clauses.

First of all, I believe the authors should talk more about their motivations. Taking Fig. 7 as an example, the authors divided the 1370km-long reach into 43 sub-reaches. From an engineering point of view, when do we need to identify river reaches like this? What is the possible application of the study? Imagining someone is working on a river restoration project; how do you convince him to use your method to segment the reach.

Second, what exactly is the author’s new approach? The author argued that the statistical approaches try to identify clusters of slices which display similar characters, while their method identifies slices belonging to diverse clusters. This statement is quite ambiguous. To use your new approach, one first need to “choose” core attributes and avoid “extensive attributes”? However, most likely we pay attention to a specific attribute when we have to segment the reach. VB width, in most cases, is the least needed information (the most accessible information as well).

Third also the fatal shortcome of the study, the authors did not explain how to segment the reach when three or more core attributes (e.g. VB width, active channel width, bed sediment texture, bed morphology, etc.) co-vary along the channel.

Based on the above concerns, I do not suggest to publish the article in the journal. 

Reviewer 4 Report

General comments: 1) Clearly identifying the river reaches is important for restoration, protection and utilization on rivers and its’ aquatic ecosystem, so that, diverse methods have been employed to identify the river reaches. In the manuscript, an approach based on the geomorphic features was applied in identification of the river reaches. However, there is a gap between the quality of this manuscript and the required scientific contribution by Water. 2) Language expression and structure of the manuscript are more like report, not scientific paper. For example, research advances (section 2) would be merged in the introduction; in the Method & data, scientific language expression would be employed in the whole manuscript, especially in the section of methods & data to descript the central part of the method such as compositive approach of chart, table and description, besides listing the items. 3) The method still requires a clearer description and expression, for example, how many attributes are comprised in the method? How to quantify the parameters in the method? How to validate the results of this approach? In addition, many figures miss some key elements that can rich the information of these figures. 4) The language needs further polishing. Special comments: 1) Line 60: the section 2 needs a scientific summary, and is merging to introduction. 2) Line 79: …a kind of intrinsic paradox…, is this paradox? Or reaches are not defined clearly here? 3) Line 87: in Fig. 1, ‘a’ and ‘b’ with black frame should be revised (a) and (b). 4) Line 196: title may be revised “Methods & data” 5) Line 235: the method needs a clear description and expression. 6) Line 402: missing the Fig. 5. 7) Line 406. Missing the Fig. 6. 8) Line 437: the essential elements are lost in the Fig. 8, for example, compass, coordinate, etc. 9) Line 445: how to deal with the scale effect? 10) Line 446: same to Fig. 8., express of the scale bar is incorrect. 11) Line 498: in the Fig. 10, what mean two arrows (blue and red)? 12) Line 501: expresses of scale should be 1:3,200,000 and 1:500,00. 13) Line 592: standard of scale express bars is required. 14) Line 564: why is approximate scale? 15) Line 709: the conclusion needs a big revise.

Round 2

Reviewer 2 Report

The most basic function of a river channel is as a conduit to transport water and sediment. As a rule, an alluvial river channel adjusts its geometry to accommodate its supplied loads of sediment and water.

Alterations in the natural flow regime of any river or stream can result in adjustments to the overall planform (geometry), cross-sectional channel geometry, bed configuration, and channel-bed slope.

Geometric adjustments may be in the form of cross-section geometry and characteristics (width, depth, roughness), planform geometry (bars, banks, sinuosity, curvature), or channel bed slope. Thus, it follows that channel geometry adjustments are the result of a dynamic interaction between imposed sediment and water loads, and constraints on adjustment, such as valley confinement, bedrock lithology, or biological effects.

The purpose of the segmentation is to characterize the types and extents of river channel and valley settings, with a greater or lesser emphasis on certain characteristics, depending on the objective in mind.

I don't see these concepts properly considered and much less satisfied with the segmentation actually proposed in this manuscript. Hence the reason for my statement "It must be kept in mind that the attributes and key factors must be valued according to the purpose of the river segmentation", which the authors seem not to have understood.

In short, this manuscript has no significant innovation and does not have relevant scientific value to the international scientific community.

Author Response

The most basic function of a river channel is as a conduit to transport water and sediment. As a rule, an alluvial river channel adjusts its geometry to accommodate its supplied loads of sediment and water.

Alterations in the natural flow regime of any river or stream can result in adjustments to the overall planform (geometry), cross-sectional channel geometry, bed configuration, and channel-bed slope.

Geometric adjustments may be in the form of cross-section geometry and characteristics (width, depth, roughness), planform geometry (bars, banks, sinuosity, curvature), or channel bed slope. Thus, it follows that channel geometry adjustments are the result of a dynamic interaction between imposed sediment and water loads, and constraints on adjustment, such as valley confinement, bedrock lithology, or biological effects.

The purpose of the segmentation is to characterize the types and extents of river channel and valley settings, with a greater or lesser emphasis on certain characteristics, depending on the objective in mind.

I don't see these concepts properly considered and much less satisfied with the segmentation actually proposed in this manuscript. Hence the reason for my statement "It must be kept in mind that the attributes and key factors must be valued according to the purpose of the river segmentation", which the authors seem not to have understood.

In short, this manuscript has no significant innovation and does not have relevant scientific value to the international scientific community.

**ANSWER: dear reviewer, we think we understand better your point now. Let us start from your last sentence with which we do not agree at all: we could not find in the literature anybody realizing the paradox we point out concerning the logical impossibility to define reaches by looking at attributes characterized by an extensive nature. This very fact, objectively, brings in a (small) scientific value. The method we developed based on this observation constitutes hence an innovation and a progress in geomorphic science because allows one to overcome this difficulty. Although this may seem a minor step, for the river community currently so engaged in exploiting the amazing capabilities of remote sensing and interpretation algorithms, this on the contrary can be considered a fundamental step because it opens the door to full automation in geomorphic characterization. As you may have noticed, however, we humbly propose it as a preliminary phase still requiring further testing; this in spite of the lengthy, careful, testing we honestly carried out on two very diverse cases.

Concerning now your point that "It must be kept in mind that the attributes and key factors must be valued according to the purpose of the river segmentation”, we think you overlooked our sentence that very clearly expresses the objective we refer to, i.e. (line 43 in the Introduction) “to identify parts of the river, clearly distinguishable from one another, to develop a meaningful description of the character and behaviour of the river and to design proper interventions.”

Reviewer 3 Report

The authors have provided a detailed answer to the reviewers’ comments one by one. At the moment I just have two minor suggestions.

First, the authors may add a few more sentences in Section 4 to explain why they chose these two cases, how representative are these two cases, and what are the implications?

Second, the current language is still unsatisfactory. The manuscript reads more like a technical report or manuals rather than a scientific paper. It would be better if the authors could refine their expression with help from some experts and merge a couple figures. For example, the authors may combine the Figs. 6, 11 and 12 and describe/compare the two cases before Section 4.1.

Author Response

Reviewer 3

The authors have provided a detailed answer to the reviewers’ comments one by one. At the moment I just have two minor suggestions.

First, the authors may add a few more sentences in Section 4 to explain why they chose these two cases, how representative are these two cases, and what are the implications?

Second, the current language is still unsatisfactory. The manuscript reads more like a technical report or manuals rather than a scientific paper. It would be better if the authors could refine their expression with help from some experts and merge a couple figures. For example, the authors may combine the Figs. 6, 11 and 12 and describe/compare the two cases before Section 4.1.

**ANSWER: yes, thank you: we now introduced a sentence explaining why we chose those rivers. And we gathered figures as you indicated.

As per English and style….we did our best; as foreigners, English is always a real problem to us. We hope you see it acceptable now.

Reviewer 4 Report

One minor suggestion in the figure 9:

1) In Figure 9., expresses of scale should be 1:3,200,000 and 1:500,00.

Author Response

Reviewer 4

One minor suggestion in the figure 9:

  • In Figure 9., expresses of scale should be 1:3,200,000 and 1:500,00.

**ANSWER: done, thank you

This manuscript is a resubmission of an earlier submission. The following is a list of the peer review reports and author responses from that submission.

Round 1

Reviewer 1 Report

The paper endeavours to propose a method for geomorphological classification of river reaches, with the technique applied to the Magdalena River, Colombia and the Baker River, Chile. However the results from these two rivers are in Appendixes (A, B) at the end of the paper. It is unusual for the analysis to be relegated to appendixes. Throughout the paper there is no explicit indication of the purpose of the classification proposed or the detailed way in which it is developed or the extent to which the results are scale dependent. The characteristics adopted in Appendixes A, B are not explained and scrutinised (e.g. bed sediment, stream character). In the classifications achieved the 43 reaches identified for the Magdalena (each averaging c.32 km long) and the 25 for the Baker (averaging 7km long) produces a result with limited application. There could be an acceptable paper from the material but it requires major revision of this text to ensure that the purpose is clearly stated, the basis for the classification is explained, and the wording throughout is clarified and made intelligible for all readers. The article published in Water on 7 April 2020 (Nani et al. 2020) explains some of the points that are not clear in this text. In revision it is necessary to show what is new in this paper (?Baker River example) and to explain what significant research advance is made beyond earlier publications.  I have indicated in general comments the places where the wording definitely needs clarification.

General comments:

Abstract requires rewriting and should indicate the purpose of the research and the way that it complements previous contributions.

Wording requires attention throughout the paper because meaning of some sentences is not sufficiently clear for the reader. e.g lines 52-55; 66-75; 77-80 (meaningful in what sense?); 84-91; 93-95 (?logical framing of the discourse?); 111-113; 115-117; 138-139; 149-152; 230-235; 242-244 (identify NOT identifying); 252-253; 302-304; 310-312; 322-326; 343-345; 361-365; 388 (….extensive attributes would result mixed up); 397-398 (?significant novelties?); 408-411; 412-415; 449-450; 452-454; 470-471 (quite alarming in some ways. ); 645-646 (while the current attributes seems quite weak (it was considered more as a trial hypothesis);

Introduction lines 41-91: the reader could expect to have a concise summary of previous approaches to river characterization followed by a clear statement of how this research develops from previous approaches. Comments about other approaches are given in Section 2 but clearer subheadings are required.

Line 241: Will meaning of A “humble” new method be clear to all readers? In what sense is humble being used?

Specific comments:

Lines 53-55: as attention is drawn to how the reach concept has not been clearly defined previously should there be a clear definition given here?

Line 102: is it resumed OR presumed?

Line 108: does this procedure refer to Wheaton et al 2015? OR is it referring to Manual segmentation? The subsections referring to the options available need to be more clearly separated.

Line 286: How are bed sediment texture, bed morphology defined and how are the values obtained? How would channel activity be characterized?

Line 338: envelop OR envelope?

Line 396: Meaning of emblematic stretches may not be clear

Line 405: lack NOT lacks?

Line 416: large flowrate rivers, is very general? Why not give discharge ranges?

Reviewer 2 Report

This manuscript examines an important topic, the objective definition of river reaches, but fails to do so in an effective manner.  Although some good points are made, such as the distinction between intensive and extensive variables and the circular logic of characterizing rivers based on reaches, which requires a characterization to define the reaches, they are lost in the poor presentation.  The paper is written in a very unusual style that ranges from conversational to convoluted that I found to be impenetrable and uninformative.  For almost every paragraph in the manuscript, I had to read and re-read several times to try to understand what the authors were trying to say, but even then their message was often lost.  The authors are not unfamiliar with the English language, but, if anything, trying to hard.  The sentences should be made more straightforward and less complex.  In addition, the colloquial tone is inappropriate for a scientific article.  Other issues include the lack of a clear description of the actual methodology and references to several unpublished works by the authors for key aspects of the study.  The two examples were placed in an appendix and failed to effectively illustrate the approach.  In summary, I don't feel like I learned anything from the paper, or even could learn anything the way it was written and presented, so in my opinion the manuscript cannot be published in anywhere near its present form.

Reviewer 3 Report

Greetings authors,

Thanks for the opportunity to review your manuscript.

I will not waste a lot of our very limited time. But I was rather disappointed by the structure, the written expression and the format of this manuscript. I can see what you were trying to say but how and what you delivered were of concern. 

I have attached the edited manuscript file with my comments included.

The overall concerns were that:

  • I found the title very misleading as I do not believe that you have delivered what you set out to do! (See below).
  • You need to write your prose using a clear and logical structure. I found your writing style very convoluted and difficult to follow.
  • Your manuscript read more like a thesis chapter than a manuscript for a journal!
  • I also felt that you were providing a rather personalised and subjective discussion within the literature review. Remember that you are supposed to present what is known on the subject in an objective and non-subjective manner. Once you have established the rigour of your research data/results, then you can challenge the works of others.  
  • You need to learn how to use Appendices. The focus of this manuscript was supposed to be on: "Geomorphic river characterization: a systematic, automated approach for river segmentation illustrated for the Magdalena River (Colombia) and the Baker River (Chile)" ... BUT the emphasis upon the Magdalena River (Colombia) and the Baker River (Chile) were non-existent in the MAIN BODY of the manuscript! In other words, if you put important parts of your manuscript in an Appendix it will NOT be read as part of the main story.
    • Therefore, one could suggest that you were actually trying to cheat on word limits by placing your case studies in an Appendix.
    • Just remember...the main story MUST stand on their own between the Abstract and the References NOT in Appendices.
    • The Appendices is normally where raw data or additional data is presented ... but that main text is well argued without the need to visit an Appendix. 

As a result of these points, I am recommending the rejection of this manuscript. 

I am sorry to be so blunt but it is important you learn how to best manage and address any future piece of writing to ensure we are all not wasting our valuable time. 

Best wishes in your future research publications.
